# Molecular basis of AKAP79 regulation by calmodulin

Neha Patel[1], Florian Stengel[2,3], Ruedi Aebersold [3,4] & Matthew G. Gold [1]

AKAP79/150 is essential for coordinating second messenger-responsive enzymes in processes including synaptic long-term depression. $Ca^{2+}$ directly regulates AKAP79 through its effector calmodulin (CaM), but the molecular basis of this regulation was previously unknown. Here, we report that CaM recognizes a '1-4-7-8' pattern of hydrophobic amino acids starting at Trp79 in AKAP79. Cross-linking coupled to mass spectrometry assisted mapping of the interaction site. Removal of the CaM-binding sequence in AKAP79 prevents formation of a $Ca^{2+}$-sensitive interface between AKAP79 and calcineurin, and increases resting cellular PKA phosphorylation. We determined a crystal structure of CaM bound to a peptide encompassing its binding site in AKAP79. CaM adopts a highly compact conformation in which its open $Ca^{2+}$-activated C-lobe and closed N-lobe cooperate to recognize a mixed $\alpha/3_{10}$ helix in AKAP79. The structure guided a bioinformatic screen to identify potential sites in other proteins that may employ similar motifs for interaction with CaM.

[1] Department of Neuroscience, Physiology & Pharmacology, University College London, Gower Street, London, WC1E 6BT, UK. [2] Department of Biology, University of Konstanz, Universitätsstrasse 10, 78457 Konstanz, Germany. [3] Department of Biology, Institute of Molecular Systems Biology, ETH Zürich, 8093 Zürich, Switzerland. [4] Faculty of Science, University of Zürich, Zürich, Switzerland. Correspondence and requests for materials should be addressed to M.G.G. (email: m.gold@ucl.ac.uk)

Calmodulin (CaM) is a $Ca^{2+}$-sensing protein that is expressed in all eukaryotic cells. It mediates many essential processes driven by $Ca^{2+}$, including long-term changes in synaptic connections in the brain[1], apoptosis[2], and immune responses[3]. CaM is a bilobal protein, with each lobe consisting of two EF hands that adopt an open conformation upon binding $Ca^{2+}$. $Ca^{2+}$/CaM binds and modulates a wide range of proteins. For example, $Ca^{2+}$/CaM-binding relieves autoinhibition in the phosphatase calcineurin and in $Ca^{2+}$/CaM-dependent protein kinase II (CaMKII). CaM possesses a highly flexible structure that is capable of recognizing a diversity of interaction motifs. Many recognition motifs for $Ca^{2+}$/CaM fall within two major classes[4]. Proteins including myosin light chain kinase[5] and calcineurin present amphiphilic helices with anchor hydrophobic residues at positions 1, 8, and 14, whereas proteins including CaMKII[6] bind CaM through shorter helical sequences with the hydrophobic

anchors at position 1, 5, and 10. High-resolution crystal and NMR structures have established how both lobes of CaM act in concert to coordinate either 1-5-10 or 1-8-14 class motifs[7,8]. Outside of these predominant classes, in a few cases $Ca^{2+}$/CaM binds in an extended conformation to longer recognition sequences[9]. For example, Munc13 presents a 1-5-8-26 pattern of hydrophobic amino acids for interaction with $Ca^{2+}$/CaM[10]. Prior to this study, in all cases where CaM binding depends on $Ca^{2+}$, all four EF hands have been found to bind $Ca^{2+}$ in high-resolution structures. In addition, with a single exception[11], the terminal hydrophobic anchor positions are separated by no fewer than eight amino acids. Despite decades of progress in understanding CaM recognition sequences, CaM-binding sites in some $Ca^{2+}$/CaM-regulated proteins have eluded mapping. One such protein is AKAP79, which was the focus of this investigation.

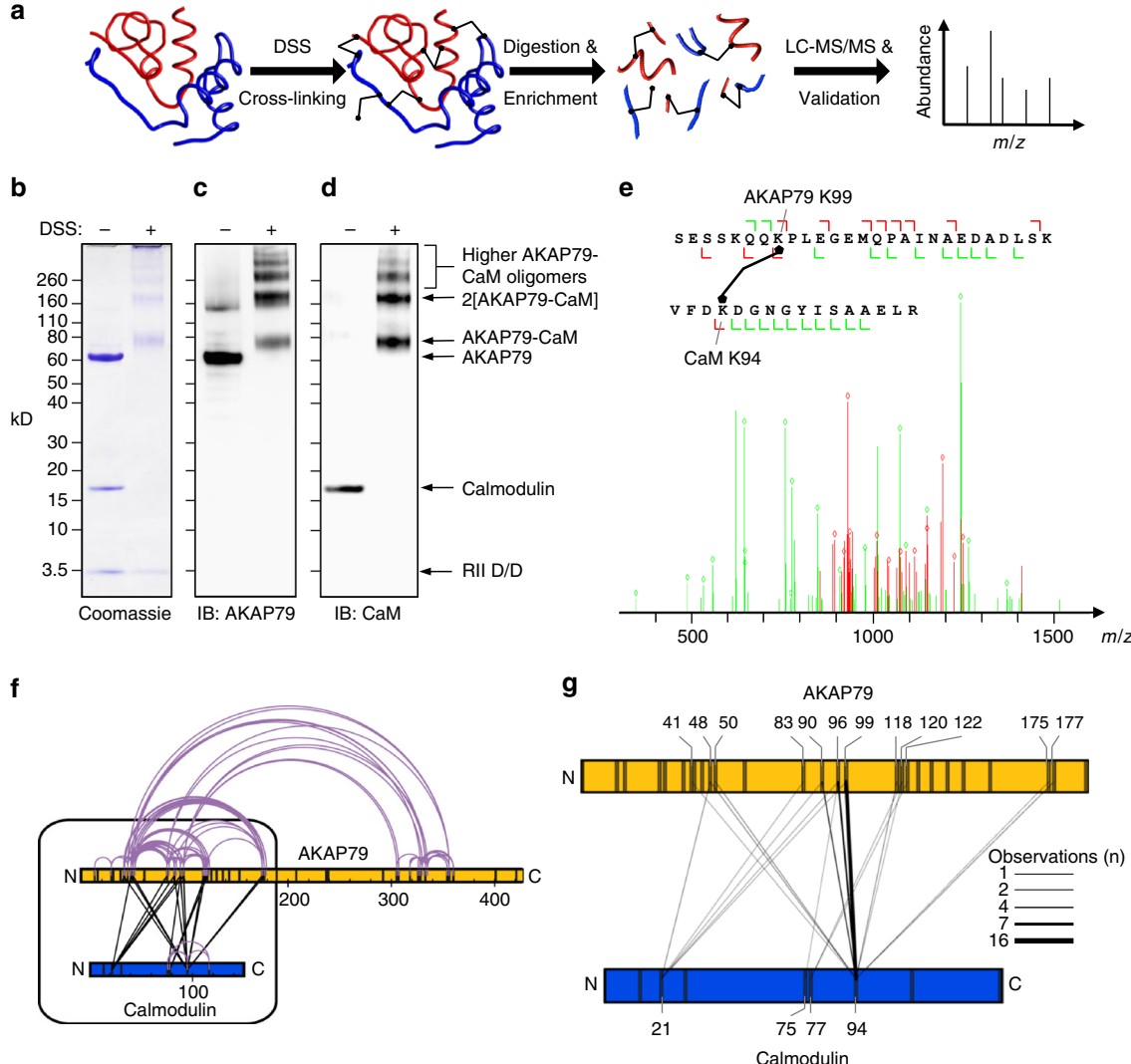

**Fig. 1** XL-MS of AKAP79 and CaM. **a** Schematic showing the workflow of XL-MS. Purified protein complexes are cross-linked under native conditions using an isotope-coded crosslinker. After trypsinization, potential cross-linked peptides are enriched by SEC and subsequently subjected to LC-MS/MS analysis. Cross-linked peptides are identified, sequenced and validated using the "xQuest/xProphet" pipeline. An equimolar mixture of AKAP79-D/D and CaM was separated by gel electrophoresis either before or after cross-linking and proteins were imaged by coomassie staning (**b**), or immunoblotting for AKAP79 (**c**) or CaM (**d**). **e** Exemplar MS/MS spectrum showing excellent agreement between experimental and theoretical spectra for an interlink between $AKAP79_{K99}$ and $CaM_{K94}$. Matches (diamonds) are indicated with a mass tolerance of 0.2 Da for common ions (*green*) and 0.3 Da for cross-link ions (*red*). **f** Distribution of interlinks (black) and intralinks (grey) detected in the cross-linked AKAP79-D/D-CaM sample. **g** Focused illustration showing distribution of interlinks between AKAP79 and CaM. Line thickness is proportional to the number of times an interlink was observed. The source data for (**f**) and (**g**) are listed in Supplementary Table 1

AKAP79 (rodent ortholog AKAP150, gene name AKAP5) is a prototypic A-kinase-anchoring protein (AKAP) that fulfills key physiological roles, including supporting long-term synaptic depression[12], directing calcineurin for NFAT dephosphorylation[13], and controling release and response to insulin[14]. AKAP79 engages in multiple protein–protein interactions, including with cAMP-dependent protein kinase (PKA)[15], calcineurin[16] and Ca$^{2+}$/CaM[17]. The calcineurin-binding site on AKAP79 has been proposed as a drug target[14,18] for treating diabetes[14,18]. It also interacts with the membrane bilayer through a combination of N-terminal polybasic regions[19] and double palmitoylation[20,21]. Constitutive binding sites for calcineurin and PKA have been mapped within AKAP79: PKA binds to an amphipathic helix near to the anchoring protein's C-terminus[15], whereas calcineurin binds to the sequence PIAIIITD between AKAP79 residues 337 and 343[19]. Initial mapping of these two anchoring sites has been corroborated by high-resolution structures[22–24], leading to functional investigations using AKAP79 variants containing targeted ablations[12,14]. Binding of Ca$^{2+}$/CaM within the first 153 amino acids of AKAP79[25] antagonizes interactions between the tandem basic regions of the protein and acidic headgroups of the membrane bilayer[19]. Ca$^{2+}$/CaM also stabilizes interactions between AKAP79 and calcineurin by triggering a secondary interface with the phosphatase that involves elements in the N-terminus of AKAP79[25], although it is not clear how CaM triggers this interface. AKAP79 contains no sequences conforming to 1-5-10 or 1-8-14 motifs, and the absence of a precise location for CaM binding has impeded progress in understanding how CaM regulates AKAP79.

We set out to determine the CaM-binding site in AKAP79 by exploiting the rapidly developing technique cross-linking coupled to mass spectrometry (XL-MS). The general approach is to cross-link proteins under native conditions, followed by detection and exact sequence identification of cross-linked peptides by MS (Fig. 1a)[26]. The interlinking pattern between CaM and AKAP79 served as a foundation for identifying and characterizing the CaM-binding site in AKAP79.

## Results

**Analysis of cross-linking CaM in complex with AKAP79.** We first purified AKAP79 bound to the dimerization and docking (D/D) domain of type IIα PKA regulatory subunits (Supplementary Fig. 1a, b). This complex, 'AKAP79-D/D', was mixed with purified CaM in an equimolar ratio and incubated with the homo-bifunctional amine-reactive cross-linker disuccinimidyl suberate (DSS) in the presence of 100 μM Ca$^{2+}$. Coomassie staining (Fig. 1b) and immunoblotting against AKAP79 (Fig. 1c) or CaM (Fig. 1d) after electrophoresis on denaturing gels confirmed that cross-linking covalently coupled CaM and AKAP79. The most prominent band at 80 kD corresponds to cross-linked AKAP79-CaM heterodimers, whereas the band at 160 kD represents stabilization of 2AKAP79-2CaM tetramers, which is consistent with native MS and gel filtration studies, indicating that AKAP79 dimerizes[25,27]. To determine which lysines had cross-linked within the complex, the sample was digested with trypsin, and cross-linked peptides were identified using liquid chromatography (LC)-MS/MS. A total of 213 peptides were identified from three biological replicates, each shot in technical duplicates. These consist of 47 interlinks between CaM and AKAP79, 148 links between AKAP79 peptides (intralinks or links between homodimeric AKAP proteins), 13 CaM-CaM intralinks, and 5 links between symmetrical peptides, indicating interlinks between AKAP79 homodimers (Supplementary Table 1). Such symmetrical dimeric interlinks were detected at three positions in AKAP79 (83-83, 50-50, 48-48), which is consistent with previous

reports that AKAP79 dimerizes in a parallel conformation involving N-terminal contacts[25]. An exemplar MS/MS sequencing spectrum is shown for a peptide linked between K94$_{CaM}$ and K99$_{AKAP79}$ (Fig. 1e). The overall pattern of intralinks (grey) and interlinks (black) within and between AKAP79 and CaM is illustrated in Fig. 1f.

Interlinks between AKAP79 and CaM were restricted to lysines between residues 41 and 177 in the N-terminus of AKAP79 (Fig. 1g)[25]. To narrow down the location of the CaM-binding site on AKAP79, we considered the frequency with which particular lysine pairs were observed in cross-linked peptides. Multiple detections of a particular link may arise as enzymatic digestion may lead to peptides of varying lengths containing the same cross-linking sites. In addition, the data set contains the data pooled from biological replicates (potential additional multiple identifications of cross-linked peptides originating from different charge states or technical duplicates were counted as the same unique cross-linked peptide). A high frequency of a particular lysine pairing therefore strongly indicates that the two amino acids are likely to be optimally positioned for cross-linking under native conditions. In this case, 66% of all interlinks were mediated between K94 in CaM (K94$_{CaM}$) and either K90$_{AKAP79}$ (4 detections, Fig. 1g), K96$_{AKAP79}$ (7 detections, Fig. 1g) or K99$_{AKAP79}$ (16 detections, Fig. 1g). The identification of this cross-linking hot-spot provided us with a reference point to triangulate the position of the CaM-binding site in AKAP79.

**Location of the CaM-binding site in AKAP79.** The cross-linking data suggested that CaM binds in the vicinity of AKAP79 lysines 90, 96, and 99. AKAP79 contains a predicted helical region between amino acids 79–86 in the vicinity of this cross-linking hot spot. To examine whether this could correspond to the CaM-binding site in AKAP79, we used a complementary method - the amplified luminescent proximity homogenous assay (alpha)-screen technique-to scan the N-terminus of AKAP79 for sequences that could mediate interactions with CaM. Incubation of GST-AKAP79 (1–153), but not control GST, with biotinylated CaM brings streptavidin donor and anti-GST acceptor beads within 200 Å, which can be detected by measuring emission between 520 and 620 nm following illumination at 680 nm (Fig. 2a, b). Ca$^{2+}$triggered CaM association with GST-AKAP79 (1-153) but not with GST alone (lanes 2 and 4, Fig. 2b, $p = 2.2 \times 10^{-8}$). We screened peptides derived from the N-terminus of AKAP79 for their ability to reduce alphascreen emission. Twenty 20-mer peptides were synthesized corresponding to a walk in increments of 5 amino acids along the N-terminus of AKAP79 beginning at 21–40 (peptide 'A', Fig. 2c) and ending at 116–135 (peptide 'T', Fig. 2c). The effect of each peptide at 100 nM concentration on CaM–AKAP79 (1–153) interaction is shown in Fig. 2d as log2 emission ratios (with/without peptide). Consistent with the cross-linking data, the only two peptides encompassing residues 79 and 86 most effectively inhibited the interaction. Peptide M (positions 71–90) reduced the signal to $0.13 \pm 0.02$ compared to untreated samples; and peptide N (76–95) reduced the signal to $0.21 \pm 0.03$ of the untreated emission. No other peptides reduced emission below 0.60 relative to untreated samples. There is one other predicted helical sequence in the AKAP79 N-terminus, which falls between residues 33 and 48, but no peptides spanning this region produced a marked reduction in alphascreen signal (peptides C-F, Fig. 2d).

To verify that CaM binds to the helix spanning residues 79–86, we first performed pull-down experiments using lysates from human embryonic kidney 293T (HEK293T) cells expressing full-length FLAG-tagged AKAP79 variants (Fig. 3a). Wild-type AKAP79 associates with both CaM sepharose, and cAMP agarose

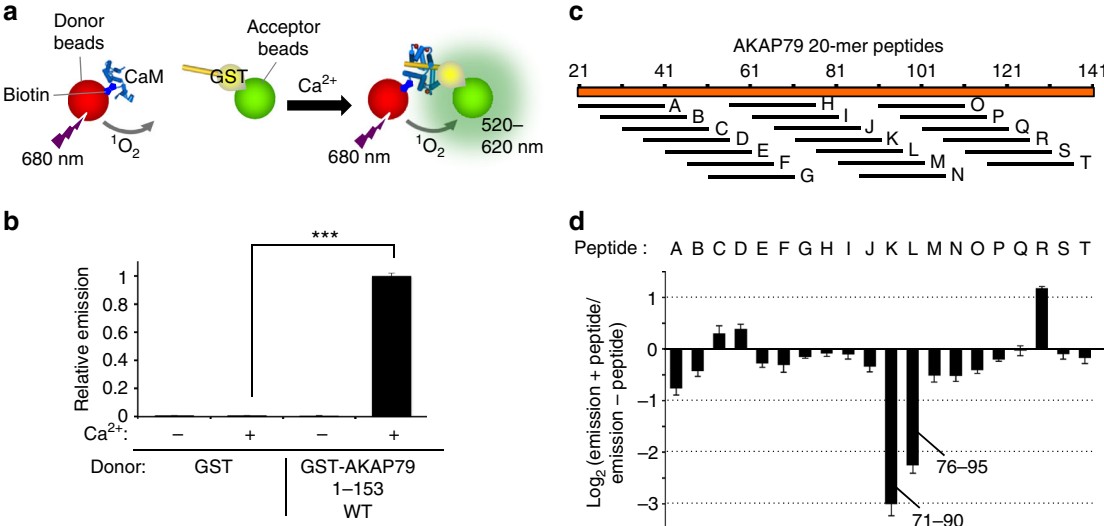

**Fig. 2** Mapping the CaM interaction in AKAP79 using an alphascreen—peptide scanning procedure. **a** Principle of the alphascreen assay. In this case, illumination at 680 nm leads to emission between 520 and 620 nm if donor and acceptor beads are co-localized through interaction of GST-AKAP79 (1–153) and biotin-CaM. **b** Alphascreen recordings of biotin-CaM association with GST or GST-AKAP79 (1–153). Recordings were performed either with or without $Ca^{2+}$ (n = 6). **c** Position of 20-mer peptides in the N-terminus of AKAP79 used for scanning. **d** Changes in alphascreen signal between biotin-CaM and GST-AKAP79 (1–153) are shown upon inclusion of each of the 20-mer peptides outlined in panel **c**. Peptides were added at 100 nM (n = 5). Error bars show s.e.m.

by virtue of its ability to interact with type II regulatory (RII) subunits of PKA (lane 1, Fig. 3a). Control experiments confirmed that CaM sepharose pull-down of WT AKAP79 was $Ca^{2+}$-dependent (Supplementary Fig. 2b). Deletion of the AKAP79 anchoring helix (Δ391-400) prevents its association with cAMP agarose but does not affect binding to CaM sepharose (lane 4, Fig. 3a), as expected. We tested two novel deletion mutants of AKAP79 lacking one or other of the two predicted helical sequences. Removal of residues 33–48 did not affect either CaM sepharose or cAMP agarose binding (lane 2, Fig. 3a). However, deletion of residues 79–86 abolished binding to CaM sepharose but not cAMP agarose binding (lane 3, Fig. 3a), which is consistent with the results of cross-linking (Fig. 1g) and peptide scanning (Fig. 2d). Removal of residues 79–86 also abolished interaction between GST-AKAP79 (1–153) and biotinylated CaM, according to the alphascreen assay (Supplementary Fig. 2c).

The predicted helical sequence spanning AKAP79 residues 79–86 is shown in Fig. 3b. The helix contains hydrophobic amino acids in a 1-4-7-8 pattern (yellow, Fig. 3b) that does not conform to conventional 1-5-10 and 1-8-14 interaction motifs. Several basic amino acids lie at the immediate C-terminus of this hydrophobic cluster, and these are also highly conserved across the AKAP5 gene family (sequence LOGO, Fig. 3b). To determine whether these basic amino acids contribute to the interaction, we compared how effectively peptides of different lengths could disrupt CaM–AKAP79 interaction. We calculated inhibition constants ($K_i$–the concentration for half-maximal inhibition) for each peptide after incubating at different concentrations with GST-AKAP79 (1–153)/biotin-CaM and measuring alphascreen emission. The peptides analyzed are shown at the bottom of Fig. 3b. Whereas a 9-mer peptide spanning positions 1–9 exhibited low efficacy with a $K_i = 5.1 \pm 1.2$ μM (black line, Fig. 3c), addition of the conserved amino acids G77 and A78 at the N-terminus brought the peptide efficacy into the nanomolar range ($K_i = 170 \pm 40$ nM, red line, Fig. 3c). Extension to a 16-mer by inclusion of the polybasic sequence 'RRKRS' increased potency further with $K_i = 75 \pm 8$ nM (black line, Fig. 3d), suggesting that these basic amino acids are involved in the interaction. Extension to 20-mer (red line, Fig. 3d, $K_i = 32 \pm 2$ nM) led to a similar fold

increase in potency, but additional extension to the 26-mer produced little further improvement (black line, Fig. 3e, $K_i = 22 \pm 1$).

Some CaM targets, such as Munc13, present a major helical hydrophobic sequence with a shorter hydrophobic motif some distance downstream—at position 26 relative to the first anchor position[10] in the case of Munc13. The primary sequence of AKAP79 does not reveal any strong candidates for a second hydrophobic motif but we nevertheless tested whether mutating the best candidate, L101, affects the potency of the 26-mer peptide. L101A substitution led to no significant reduction in potency ($K_i = 24 \pm 2$, red line, Fig. 3e). Overall the cross-linking, pull-down and alphascreen experiments show that AKAP79 contains an unconventional 1-4-7-8 motif beginning at W79 that is principally responsible for $Ca^{2+}$-dependent CaM binding, supported by a stretch of ~10 basic-rich amino acids following the motif.

**CaM-dependence of interaction between calcineurin and AKAP79.** Having established the location of the CaM-binding site in AKAP79, we next explored how the CaM-binding sites in AKAP79 and calcineurin contribute to $Ca^{2+}$/CaM-dependent interaction between the N-terminus of AKAP79 and calcineurin. CaM could conceivably drive these interactions by binding either or both of its interaction sites in calcineurin and AKAP79. We first engineered two point mutations into calcineurin (I396A/I400A) that CaM-calcineurin peptide crystal structures[28] indicate would render the phosphatase deficient in binding CaM (Fig. 4a). Assays of phosphatase activity against a phosphorylated peptide confirmed that the I396A/I400A calcineurin variant is activated with a half-maximal CaM concentration ($K_{CaM}$) >20-fold higher ($k = 670 \pm 60$ nM, red in Fig. 4b) than the WT phosphatase ($K_{CaM} = 30 \pm 12$ nM, black in Fig. 4b). Further details of curve fitting to determine these values are provided in Supplementary fig. 3. Next, we performed calcineurin pull-downs using GST-fusions to the N-terminus (1–153) of AKAP79 (Fig. 4c) in the presence of $Ca^{2+}$ and 100 nM CaM. Removing the 1-4-7-8 motif in AKAP79 (Δ79-86) led to marked reductions in pull down of

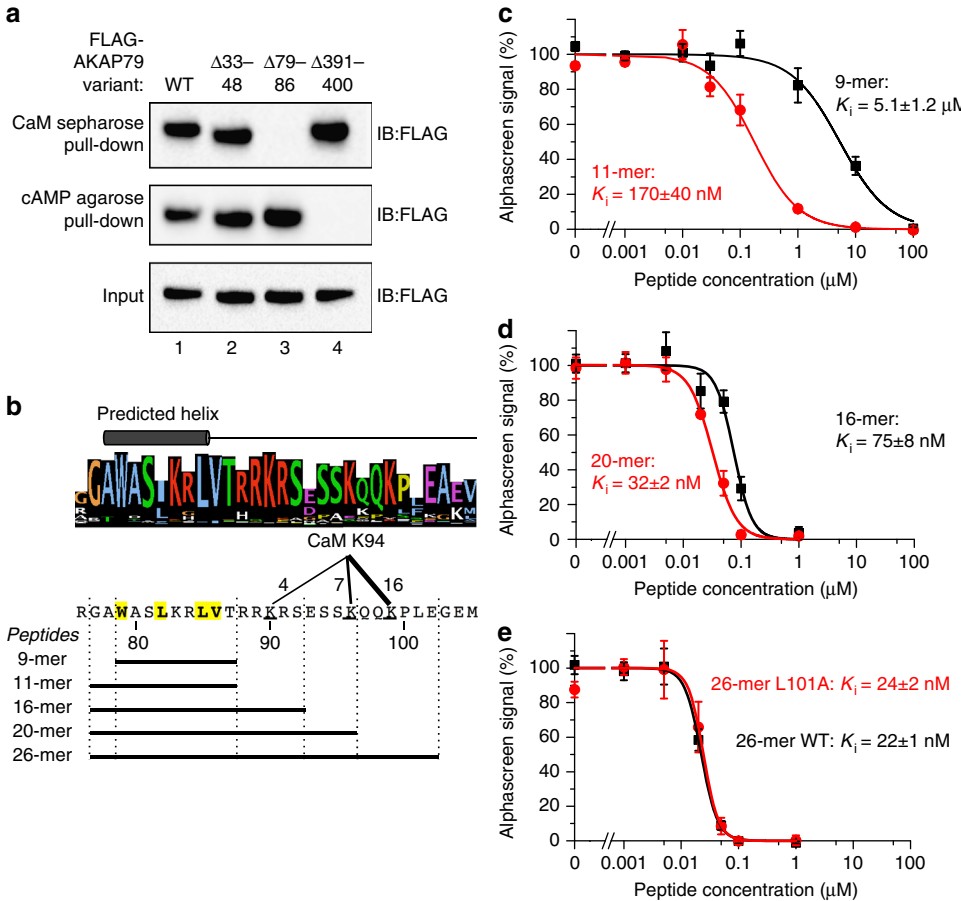

**Fig. 3** Delineation of key residues in AKAP79 required for CaM binding. **a** Pull-down of either WT, Δ33–48, Δ79–86, or Δ391–400 FLAG-tagged-AKAP79 (inputs shown in bottom panel) with either CaM sepharose (top panel) or cAMP agarose (middle panel). AKAP79 was detected by anti-FLAG immunoblotting. The experiment was performed in triplicate with each replicate producing the same pattern of bands. **b** Sequence LOGO for AKAP5 gene products aligned with predicted helical region. The cross-linking cluster between AKAP79 positions 90–99 and K94 in CaM is indicated along with the boundaries of peptides used in the following panels. **c–e** Determination of inhibitory constants for the peptides outlined in (**b**) in disrupting interaction between biotin-CaM and GST-AKAP79 (1–153) detected using the alphascreen assay ($n = 4$ for all data points). $K_i$ constants were determined for the 9-mer and 11-mer peptides **c**, 16-mer and 20-mer peptides **d**, and for either WT or L101A 26-mer peptides **e**.***$P < 0.001$ by two-tailed Student's t-test. Error bars show s.e.m.

either WT (lanes 2 & 3, Fig. 4c) or I396A/I400A calcineurin (lanes 5 & 6, Fig. 4c). Densitometry indicated a ~10-fold reduction in anti-calcineurin immunoblot signal upon removal of the 1-4-7-8 motif (Fig. 4c, $n = 3$, $p = 0.0097$). Conversely, phosphatase pull-down using GST-WT AKAP79 (1–153) was hardly altered when WT calcineurin was substituted with the I396A/I400A variant (lanes 2 & 5, Fig. 4c). These experiments suggest that the CaM-binding site in AKAP79 is required for driving the $Ca^{2+}$-dependent calcineurin-AKAP79 interface, whereas calcineurin activation is not required. It is important to note that these experiments do not rule out the possibility that CaM binding to the phosphatase is required since this occurs at markedly lower concentrations than are required to activate the phosphatase[29]. We also examined the effect of removing the 1-4-7-8 motif in live HEK293T cells using A-Kinase Activity Reporter 4 (AKAR4)[30]. The FRET efficiency of AKAR4 increases upon phosphorylation of a central phosphorylation site such that the emission ratio at 535/485 nm presents a real-time readout of phosphorylation at a PKA site (Fig. 4d). We found that the baseline FRET ratio was increased by $8.1 \pm 1.4$ %—indicative of higher resting PKA phosphorylation—when the cells were co-transfected with Δ79–86 AKAP79 rather than WT AKAP79 (Fig. 4e, $n = 12$, $p = 8 \times 10^{-5}$). In sum, the experiments outlined in Fig. 4 show that

removing the CaM-binding site in AKAP79 has consequences for PKA phosphorylation of cellular substrates.

**Structure of CaM bound to its AKAP79 recognition site**. To understand how the atypical '1-4-7-8' motif in AKAP79 is able to bind CaM, we attempted to co-crystallize CaM with peptides spanning the motif. Co-crystallization was achieved with a peptide corresponding to AKAP79 amino acids 77–92. X-ray diffraction images were collected to a resolution of 1.7 Å enabling determination of the crystal structure by molecular replacement using the C-lobe of CaM in complex with MARCKS peptide[11]. The structure was autobuilt using Buccaneer[31], and refined in Coot[32] and PHENIX[33]. The assymetric unit contains two copies of CaM–AKAP79 peptide complex (Supplementary Fig. 4a). For each copy, AKAP79 amino acids 78–88 are visible in the electron density (Supplementary Fig. 4b). The two copies of the complex are similar with overall root mean standard deviation (rmsd) for Cα atoms = 0.834 Å. They show closer similarity between their CaM C-lobes (amino acids 80–145, rmsd = 0.276 Å), and in the AKAP79 peptide (0.597 Å), than between N-lobes (amino acids 2–79, rmsd = 0.87 Å). Unexpectedly, while all of the C-lobe EF hands coordinate $Ca^{2+}$ ions in the open conformation (light blue,

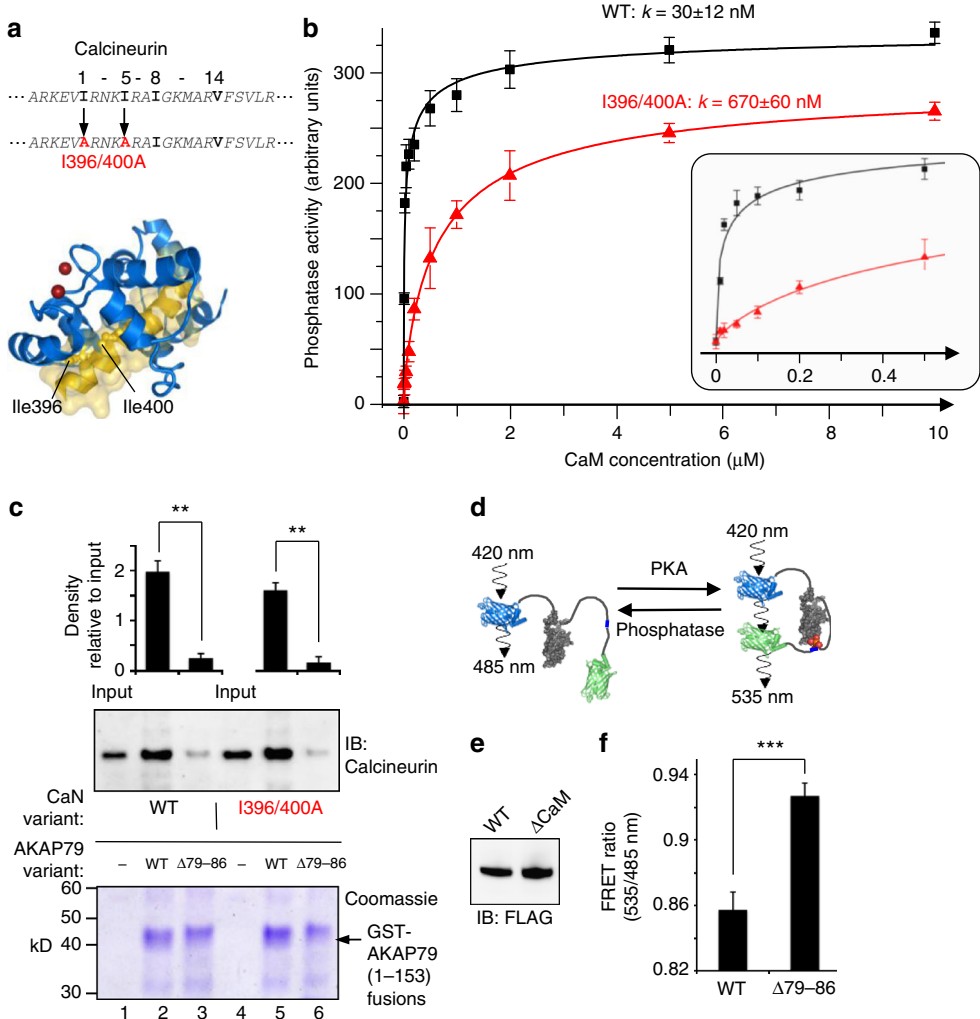

**Fig. 4** The 1-4-7-8 site is required for $Ca^{2+}$-sensitive interactions with calcineurin. **a** Location of two isoleucine to alanine mutations in the sequence and structure of calcineurin that render the phosphatase less responsive to CaM. **b** Phosphatase activity of WT and I396A/I400A calcineurin as a function of CaM concentration. Inset: activity between 0–0.5 μM CaM. **c** WT or I396A/I400A calcineurin pull-down with either WT or Δ79-86 GST-AKAP79 (1–153). Comparisons were performed using densitometry for anti-calcineurin immunoblots ($n = 3$). **d** Principle of PKA activity recordings using AKAR4: phosphorylation within an FHA recognition site triggers a structural rearrangement that increases FRET efficiency between terminal fluorescent proteins. **e** Anti-FLAG immunoblot of lysates from HEK293T cells used in AKAR4 recordings showing relative expression levels of WT and Δ79-86 FLAG-AKAP79. **f** Resting FRET ratios of cells transfected with AKAR4 and either WT or Δ79-86 FLAG-AKAP79 ($n = 12$). **$P < 0.01$ by two-tailed Student's t test. Error bars show s.e.m.

Fig. 5a, b), the N-lobe EF hands are in a closed metal ion-free conformation (dark blue, Fig. 5a, b). Consistent with the absence of $Ca^{2+}$ ions in the N-lobes, CaM amino acids 56–59 (part of the second N-lobe EF-hand) are not visible in the electron density.

Although only the C-lobes coordinate $Ca^{2+}$, remarkably both CaM lobes engage the AKAP79 peptide (orange, Fig. 5b). All four hydrophobic side-chains of the AKAP79 1-4-7-8 motif are accommodated within a hydrophobic pocket formed by the coming together of both CaM lobes (Fig. 5b). Trp79 in AKAP79 ($W79_{AKAP79}$) is deeply buried in a cleft within the CaM C-lobe, which is the canonical binding position for the first hydrophobic anchor amino acid. To determine the relative importance of different amino acids within the 1-4-7-8 motif for CaM association, we compared the ability of mutant peptides derived from AKAP79 77–96 to disrupt interactions between biotin-CaM and GST-AKAP79 (1–153). Whereas wild-type peptide reduced emission to $0.03 \pm 0.01$ relative to untreated samples, substitution of Trp79 with alanine greatly reduced the potency of the peptide ($p = 0.0013$) with a small reduction in emission to $0.78 \pm 0.06$

relative to the untreated condition (Fig. 5c). The substitution V86A slightly reduced potency ($0.17 \pm 0.04$ remaining emission, $p = 0.019$, compared to the WT peptide), while the helix-destabilizing mutation L82P also markedly reduced the potency of the peptide ($0.61 \pm 0.08$ relative emission, $p = 0.0034$, Fig. 5c). Alanine substitutions at positions '4' (L82) and '7' (L85) had no significant effect on potency, indicating that they are less essential for the interaction (lanes 3 and 4, Fig. 5c). We also translated these findings into the context of full-length AKAP79 on the basis that studies with peptides might not capture the full complexity of the interaction. CaM sepharose pull-down experiments with purified full-length AKAP79-RIIα D/D complexes support a key role for $W79_{AKAP79}$ in mediating association with CaM. Whereas wild-type AKAP79 efficiently bound to CaM sepharose (Fig. 5d), both the deletion of positions 79–86, or substitution of tryptophan 79 for alanine, abolished binding (Fig. 5d).

The binding mode is unusual following anchoring of $Trp79_{AKAP79}$ within the CaM C-lobe. Strikingly, the AKAP79 polypeptide twists away from CaM after position 86 (orange,

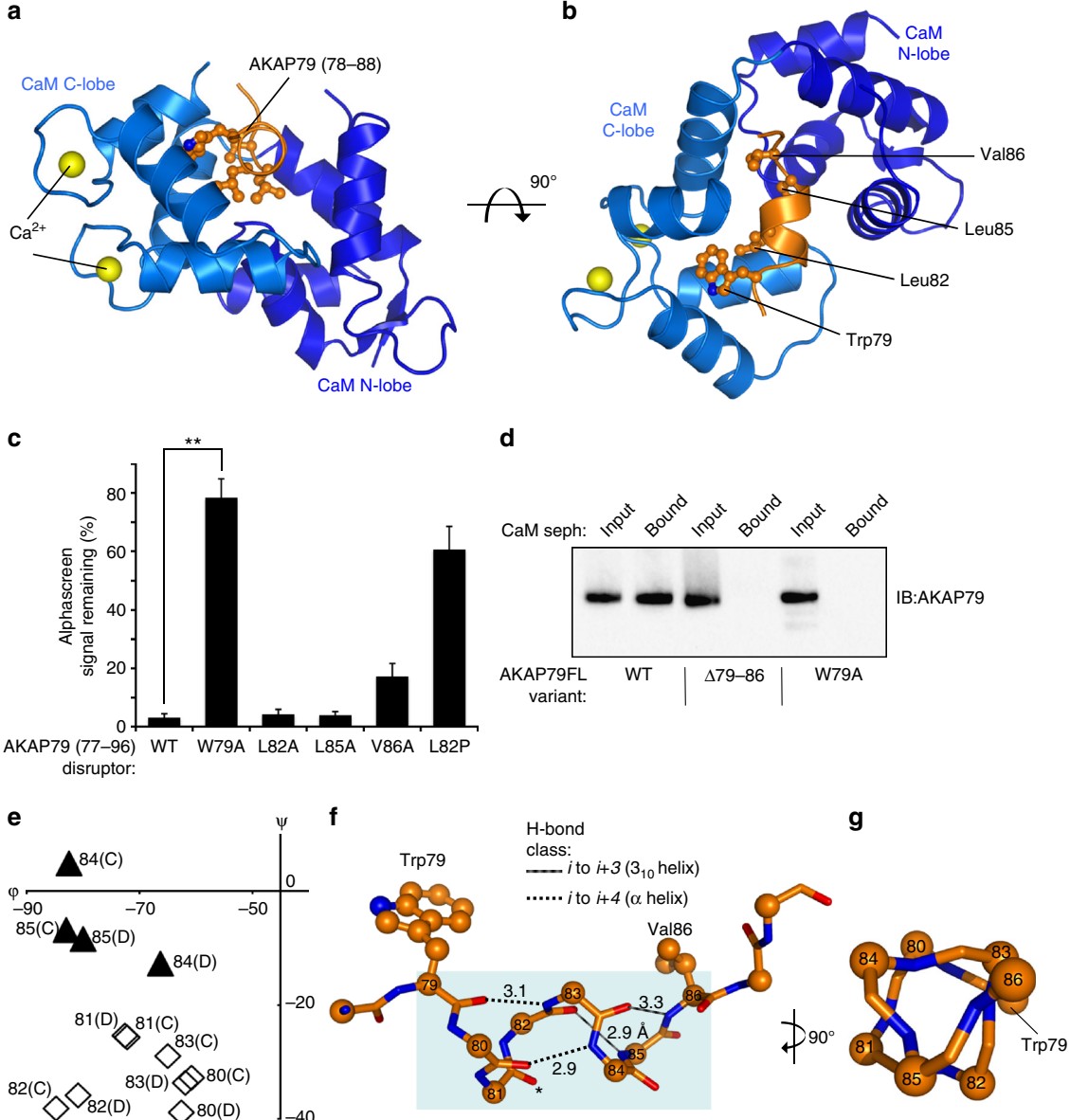

**Fig. 5** Crystal structure of CaM in complex with its AKAP79 binding site. **a** Cartoon representation showing one of the two copies (chains B and D) of AKAP79 peptide (orange) bound to CaM (*blue*) in the asymmetric unit. The C-lobe (lighter *blue*) is in the open conformation with each of its two EF hands coordinating $Ca^{2+}$ (yellow). **b** Rotation of the complex through 90° highlighting the position of the four hydrophobic amino acids comprising the 1-4-7-8 motif. **c** Reduction in alphascreen signal between biotin-CaM and GST-AKAP79 (1–153) upon addition of 20-mer peptides derived from AKAP79 77–96 (*n* = 4). The effects of point mutations within the disruptor peptide were compared. **d** Binding of purified full-length WT, Δ79–86 or W79A AKAP79 to CaM sepharose. Each AKAP79 variant was purified in complex with the D/D of RIIα. AKAP79 was released from the beads by incubation with EGTA, and detected by anti-AKAP79 IB. The experiment was performed in triplicate with each replicate leading to the same pattern of bands. **e** Limited Ramachandran plot showing dihedral angles for both copies of AKAP79 positions 80–85 in the asymmetric unit. Black triangles represent amino acids with angles characteristic of $3_{10}$ helices; white diamonds are amino acids with α-helical geometry. **f** Representation of backbone H-bonds within the AKAP79 helix with distances shown in Å. The two α-helix-type bonds are shown by dotted lines; $3_{10}$-helical H-bonds as striped lines. The carbonyl group of S81 that does not H-bond to a backbone group is asterisked. **g** Rotation of the helix through 90°. The triangular backbone geometry of positions 83–86 is such that the side-chains of W79, L83 and T86 extend in the same direction. \*\**P* < 0.01 by two-tailed Student's *t* test. Error bars show s.e.m.

Fig. 5b), whereas most CaM-interacting proteins present single straight α-helices for interaction. CaM adopts a highly compact conformation to accommodate AKAP79. We searched for structural homologs of full-length CaM taken from the complex using the DALI server and returned no structural matches spanning both lobes with rmsd < 3 Å (Supplementary Table 2). This analysis confirms that, in this crystal structure, CaM adopts a conformation that has not been observed before.

**The binding helix contains $3_{10}$ helical character**. The Ramachandran plot for amino acids in AKAP79 shows that six amino acids in the structure exhibit the geometry of right-handed helices (positions 80–85, Fig. 5e). Whereas positions 80–83 exhibit α-helical backbone geometry (diamonds, Fig. 5e), positions 84 and 85 possess more positive ψ angles that are characteristic of $3_{10}$ helices (black triangles, Fig. 5e). Analysis of the hydrogen-bonding pattern within AKAP79 confirms that the AKAP79 helix

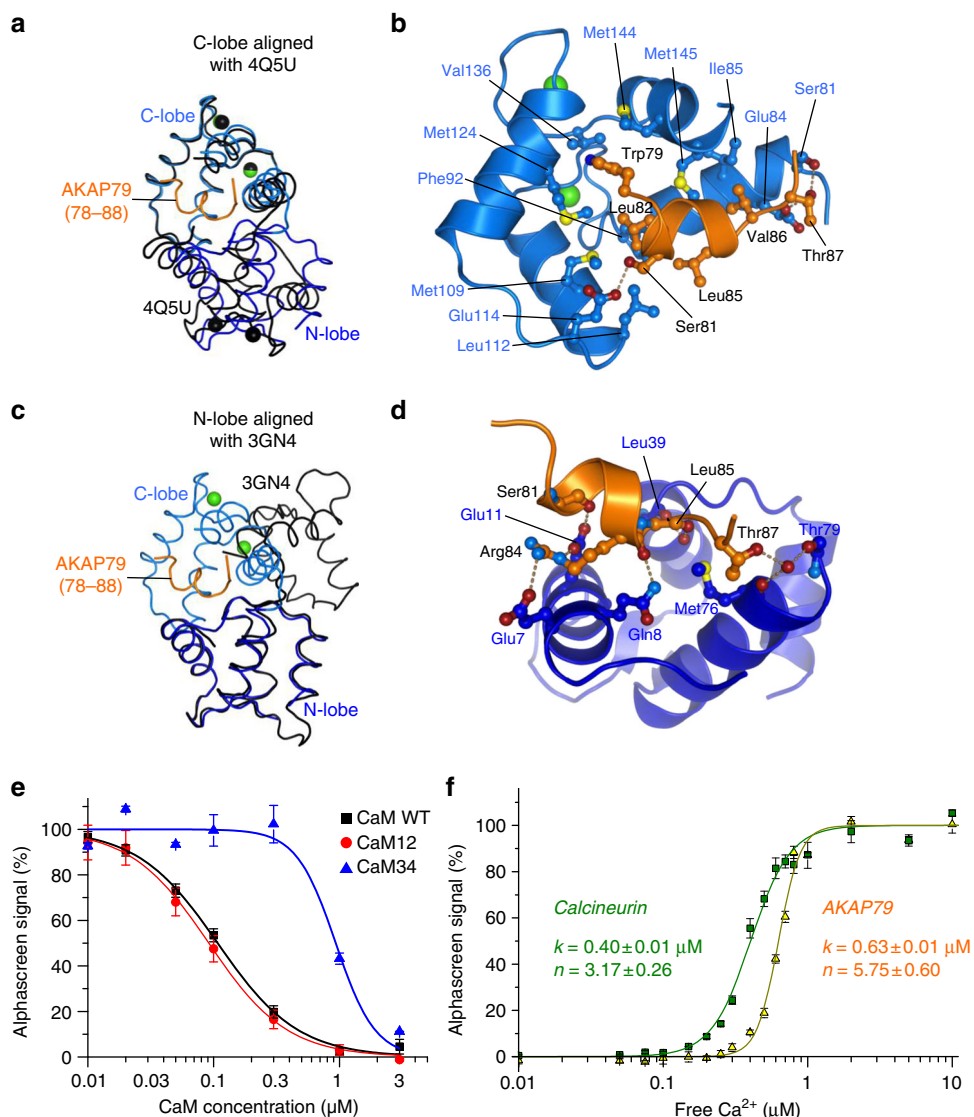

**Fig. 6** Features of N and C lobe interactions and Ca²⁺ regulation. **a** Alignment of the CaM–AKAP79 complex (blue and orange, respectively) to CaM from PDB ID 4Q5U (black). The structures are aligned through the C-lobes of CaM. **b** Close up of interactions between the CaM C-lobe (light blue) and AKAP79 (orange). H-bonds are shown as dotted lines, Ca²⁺ ions are yellow. **c** Equivalent to (**a**) but instead aligned to PDB ID 3GN4 (*black*) through the N-lobes of CaM. **d** Close up of interactions between the N-lobe (dark blue) and AKAP79 (orange) showing the network of H-bonds between the two. **e** Comparison of WT CaM (black), CaM12 (red) and CaM34 (blue)-mediated disruption of biotin-CaM and GST-AKAP79 (1–153) association, as measured by alphascreen assay (*n* = 3). **f** Binding of biotin-CaM to either GST-calcineurin (green) or GST-AKAP79 1–153 (orange) was monitored at different free Ca²⁺ concentrations (*n* = 4) using the alphascreen assay. Error bars show s.e.m.

exhibits $3_{10}$ helicity at its C-terminus. The L85$_{AKAP79}$ backbone amino group H-bonds with the carbonyl group of L82$_{AKAP79}$, and the equivalent atom in V86$_{AKAP79}$ also H-bonds with the carbonyl group of the amino acid three residues upstream (Fig. 5f, striped lines). The characteristic triangular $3_{10}$ helical conformation between amino acids 83 and 86 is most clearly visible when looking down the barrel of the AKAP79 helix (Fig. 5g). The backbone groups of pairs 79–83 and 80–84 engage in α-helical H-bonds (Fig. 5f, dotted lines). This overall helical geometry enables AKAP79 to present an amphipathic helix, in which all four hydrophobic positions, including W79$_{AKAP79}$ and V86$_{AKAP79}$ (Fig. 5f) are projected from the same side of the helix. This is the first example of a CaM-binding helix containing $3_{10}$ helical character.

**Analysis of Ca²⁺-dependence of the CaM–AKAP79 interaction.** Although there is no close structural homolog of the

conformation of full-length CaM observed in complex with AKAP79, both the N- and C-lobes separately resemble conformations observed in other crystal structures of the Ca²⁺ sensor (Supplementary Table 2). The C-lobe in particular has more than 20 structural homologs within an rmsd of 1 Å, including the C-lobe of CaM in complex with the 1-8-14 motif of calcineurin[28] (rmsd 0.5 Å, Cα positions 80–147). However, in cases where the C-lobe adopts a similar conformation, the N-lobe is always markedly different with both EF hands coordinating Ca²⁺. For example, Fig. 6a shows differences in both the conformation and position of the N-lobes of CaM in complex with AKAP79 (blue) or calcineurin (black) when the CaM C-lobes are aligned. Nevertheless, this structural analysis indicates that the C-lobe, in complex with AKAP79, adopts an essentially canonical open Ca²⁺-bound conformation that is able to accommodate a linear pattern of hydrophobic side-chains. In this case, the four hydrophobic side-chains of the amphipathic 1-4-7-8 motif engage

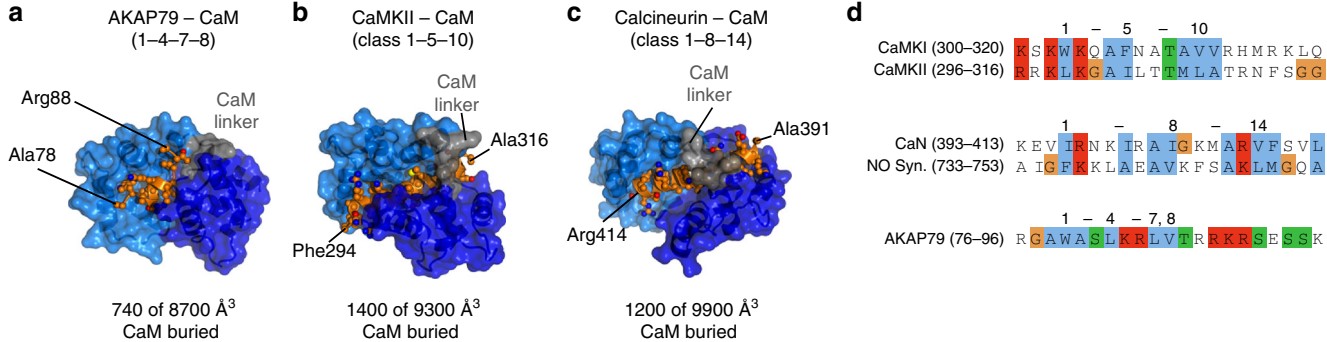

**Fig. 7** Comparison of the 1-4-7-8 sequence to previously identified CaM interaction motifs. **a–c** Representations of CaM from the same orientation in complex with three types of $Ca^{2+}$-dependent interaction motif: (**a**) 1-4-7-8 motif in AKAP79 (**b**) 1-5-10 class motif from CaMKII (PDB ID 2WEL) (**c**) 1-8-14 class motif from calcineurin (PDB ID 2JZI). **d** Sequence alignment of the AKAP79 1-4-7-8 motif with classical 1-5-10 and 1-8-14 motifs

in van der Waals contacts with this binding pocket. $W79_{AKAP79}$ is sandwiched between $M124_{CaM}$ and $M144_{CaM}$; $L82_{AKAP79}$ packs against $F92_{CaM}$; $L85_{AKAP79}$ engages the side-chains of both $F12_{CaM}$ and $L39_{CaM}$; and $V86_{AKAP79}$ contacts both $M145_{CaM}$ and $I85_{CaM}$ (Fig. 6b). The side-chains of $A78_{AKAP79}$ and $T87_{AKAP79}$ also contact hydrophobic patches on the surface of the C-lobe (Fig. 6b), whereas the side-chains of $S81_{AKAP79}$ and $T87_{AKAP79}$ form direct H-bonds with this same lobe (dotted lines, Fig. 6b).

The isolated N-lobe possesses fewer close structural homologs. The highest scoring match is to the N-lobe of CaM in complex with an IQ motif from myosin VI (rmsd 1 Å, Supplementary Table 2)–the class of motif that CaM binds in the $Ca^{2+}$-free state[34]. All of the CaM N-lobe structural homologs retrieved by DALI, including structures of apo CaM[35], are closed and $Ca^{2+}$-free (Supplementary Table 2). Although CaM also employs a closed N-lobe to engage myosin VI, the relative orientations of the N- and C-lobes are markedly different in this complex (black, Fig. 6c) compared to with AKAP79 (blue, Fig. 6c). For example, $E7_{CaM}$, $Q8_{CaM}$ and $E11_{CaM}$, which engage in H-bonds with AKAP79, are not involved in the interaction with myosin VI and instead face into solution in the CaM-myosin complex. Closer inspection of the CaM–AKAP79 peptide complex reveals that N-lobe H-bonds, involving residues including $E7_{CaM}$, $Q8_{CaM}$ and $E11_{CaM}$, enable CaM to accommodate the atypical AKAP79 helix (Fig. 6d). Notably, the side-chain of $E11_{CaM}$ H-bonds to the backbone carbonyl of $S81_{AKAP79}$, thus satisfying an H-bond lost in the transition from α to $3_{10}$ helix (marked by an asterisk in Fig. 5f). Similarly, the side-chain of $Q8_{CaM}$ fulfills a helix-capping function in H-bonding to the backbone carbonyl of $L85_{AKAP79}$ (Fig. 6d). In addition to this network of H-bonds, N-lobe residues $L39_{CaM}$ and $F12_{CaM}$ form a hydrophobic pocket that enables van der Waals interactions with the side-chain of $L85_{AKAP79}$ (Supplementary Fig. 4c).

The crystal structure suggests that $Ca^{2+}$ binding to the CaM N-lobe is not required for binding AKAP79. To test this, we compared the ability of CaM mutants unable to bind $Ca^{2+}$ in either the N-lobe (CaM12) or C-lobe (CaM34) to inhibit interaction between biotin-CaM and GST-AKAP79 (1–153). CaM12 inhibited the interaction with a similar $K_i$ ($91 \pm 6$ nM, red line, Fig. 6e) to wild-type CaM ($108 \pm 8$ nM, black line), consistent with the notion that $Ca^{2+}$ binding to the CaM N-lobe is not required for interaction with AKAP79. CaM34 ($K_i = 930 \pm 130$ nM, blue line, Fig. 6e) was a significantly less potent inhibitor of the interaction, which supports the idea that generation of the open conformation of the C-lobe by $Ca^{2+}$ binding enables interaction with AKAP79. Although $Ca^{2+}$ binding to the CaM N-lobe is apparently not required for association with AKAP79, we found that CaM still associates with AKAP79 at relatively low

$Ca^{2+}$ concentrations: alphascreen measurements show that the half-maximal concentrations of free $Ca^{2+}$ [$Ca_{50}$] for CaM binding to AKAP79 and calcineurin are similar (Fig. 6f). Biotin-CaM bound to GST-calcineurin with $Ca_{50} = 0.41 \pm 0.01$ μM (green, Fig. 6f), whereas the concentration was slightly increased to $0.64 \pm 0.01$ μM when the same experiment was performed with GST-AKAP79 (1–153) (orange, Fig. 6f). $Ca^{2+}$ binding to CaM was highly cooperative in both cases (calcineurin, $n = 3.2 \pm 0.2$; AKAP79, $n = 4.4 \pm 0.3$). The small difference in $Ca^{2+}$ sensitivity suggests that at certain intermediate cellular $Ca^{2+}$ concentrations, calcineurin may be active while in complex with AKAP79, while the CaM-binding site in the anchoring protein remains unoccupied. Overall, the structural and alphascreen data show despite an unusual binding mode, $Ca^{2+}$ sensitivity of the interaction with AKAP79 is similar to interactions with canonical CaM target sequences.

**CaM adopts a highly compact conformation to bind AKAP79.** There are striking differences between the binding mode of CaM with AKAP79 in comparison to that observed in complexes containing class 1-5-10 and class 1-8-14 sequences. In complex with AKAP79, CaM adopts a highly compact conformation (Fig. 7a) exposing only 8700 $Å^3$ surface area. This reflects the unusually short spacing[9] of 11.9 Å between Cα atoms of the terminal hydrophobic anchor positions $W79_{AKAP79}$ and $V86_{AKAP79}$. In comparison, CaM presents 9300 $Å^3$ total surface area when binding to the 1-5-10 motif of CaMKII (Fig. 7b, 14.4 Å between anchor positions), and 9900 $Å^3$ when binding the 1-8-14 motif of calcineurin (Fig. 7c, 19.5 Å between anchor positions). Another marked difference between AKAP79 and 1-5-10/1-8-14 motifs is the path of the recognition motif through CaM. The binding sequence in AKAP79 does not form a single straight helix, but twists out from CaM after $L85_{AKAP79}$ with dihedral angles characteristic of a β-sheet for positions equivalent to '9' and '10'. Concomitantly, CaM buries only 740 $Å^3$ in the inter-action with AKAP79 (Fig. 7a). When the three binding modes are viewed from the same angle looking down on the amphiphilic side of the binding helix, only the class 1-5-10 and 1-8-14 motifs pass underneath the CaM linker (grey, Fig. 7a–c). As a result, the interfaces with CaMKII (1400 $Å^3$, Fig. 7b) and calcineurin (1200 $Å^3$, Fig. 7c) are much larger.

**Discussion**

This investigation has generated multiple findings concerning the structure and function of AKAP79 regulation by CaM, and the results have wider implications for regulation by CaM. The CaM site in AKAP79 had eluded discovery for 20 years following the

initial observation that AKAP79 binds CaM[17]. Mapping the site has enabled us to perform the first experiments using CaM-binding-deficient variants of AKAP79 to establish that $Ca^{2+}$/CaM-dependent interactions between AKAP79 and calcineurin require the CaM-binding site in AKAP79. We also demonstrated that ablating the CaM-binding site in AKAP79 increases resting PKA phosphorylation in HEK293T cells. The binding mode observed between CaM and AKAP79 further expands the documented binding capabilities of CaM[9]. The CaM–AKAP79 crystal structure reveals three abilities of CaM: it can adopt an extremely compact conformation (8700 Å surface area, Fig. 7a); associate with sequences containing $3_{10}$ helical turns; and bind a target using both lobes where only the C-lobe is in the open $Ca^{2+}$-activated conformation (Fig. 5a). Overall, the study highlights the remarkable flexibility of CaM, and the utility of XL-MS for identifying protein—protein interaction sites.

Binding of $Ca^{2+}$/CaM to AKAP79 can have two effects: it can trigger an additional interface with calcineurin (Fig. 4c), and it has previously been shown to reduce binding of the anchoring protein to vesicles containing acidic phospholipids[19]. The second polybasic sequence in AKAP79 (region 'B' between positions 75–108) is particularly important for $PIP_2$ binding[19]. Six basic amino acids fall within or at the immediate C-terminus of the 1-4-7-8 motif (lysines 83 and 90; arginines 84, 88, 89 and 91) within region B. It appears likely that CaM occludes these six basic amino acids upon binding, thereby releasing the anchoring protein from acidic phospholipid headgroups. The $Ca^{2+}$/CaM-dependent calcineurin interaction site also falls within the N-terminus of AKAP79. In this study, experiments with the reporter AKAR4 indicate that removing the 1-4-7-8 motif within AKAP79 elevates resting PKA phosphorylation in HEK293T cells (Fig. 4e). However, much remains to be understood concerning the purpose of the $Ca^{2+}$/CaM-sensitive AKAP79-calcineurin interface. Comparison of knockin mice containing targeted ablations of the constitutive PKA and calcineurin anchoring sites in AKAP150 has proved to be highly valuable for disentangling the roles of these sites in insulin signaling and long-term depression[12,14]. A similar approach incorporating mice containing knockin variants of AKAP150 lacking the 1-4-7-8 motif may now be applied now the site is mapped.

We were curious to investigate whether the binding motif in AKAP79 is a one-off, or if it might represent the first example of a hitherto unappreciated class of CaM interaction motif (Fig. 7d). On the basis of the AKAP79-CaM crystal structure, we developed a position-specific scoring matrix to search the human proteome for potential 1-4-7-8 motifs with Scansite 3. The best 100 matching sequences are listed in Supplementary Table 3. The list is likely to contain many false positives but contains several sites that warrant further investigation. The list includes five potential 1-4-7-8 motifs in other AKAPs: three similar sequences within Gravin, one in BIG1, and one in BIG2. A previous study identified a ~200 amino acid region of Gravin that can bind CaM without precisely defining the binding sites[36]: all three potential 1-4-7-8 motifs (with initial tryptophan anchors at positions 610, 759, and 804) fall within this region. The search also retrieved a potential 1-4-7-8 motif present in seven of the ten voltage-dependent $Ca^{2+}$ channel pore-forming α subunits ($Ca_V1$ and $Ca_v2$-type subunits, Supplementary Table 3). The sequence falls within a long helix in a short intracellular loop between transmembrane helices S4 and S5 in domain II[37] that moves upon voltage-dependent activation[38]. Conceivably access to this site could be altered upon voltage-dependent activation of the channels, providing a mechanism for coupling $Ca^{2+}$ and depolarization to channel regulation. Additional putative 1-4-7-8 sites of note include one potential site close to the membrane-tethered C-terminus of BCL-2, a key anti-apoptotic protein[39]. It will be exciting to see which of this long list of putative sites prove to be high affinity CaM-binding sites with physiological significance.

## Methods

**Protein expression and purification.** All protein purification steps were performed at 4 °C. To express AKAP79 in bacteria in a non-aggregated form, we cloned the D/D domain (residues 1–45) of human PKA RIIα into pGEX6P1 for expression as a PreScission-cleavable fusion at the N-terminus of GST, and included a second cistron containing N-terminally 6xHis-tagged AKAP79. Primer sequences are listed in Supplementary table 4. Co-expression of AKAP79-D/D was induced in BL21 (DE3) *E. coli* (Novagen) at $OD_{600} = 0.5$ with 0.4 mM IPTG, and the cells were harvested after 3 h at 37 °C. The cell pellet was resuspended in lysis buffer (30 mM Tris-HCl pH 8, 300 mM NaCl, 10 mM imidazole, 1 mM benzamidine, one EDTA-free protease inhibitor tablet/100 mL (Roche), 0.1 mg mL$^{-1}$ lysozyme), sonicated briefly, and clarified by centrifugation at $40,000 \times g$ for 30 min. The supernatant was collected and incubated with Ni-NTA agarose (Qiagen) for 2 h prior to elution in 30 mM Tris-HCl pH 7.5, 300 mM NaCl, 300 mM imidazole, 1 mM benzamidine. The eluted protein was buffer exchanged into glutathione sepharose binding buffer (25 mM Tris pH 7.4, 500 mM NaCl, 2 mM DTT, 1 mM EDTA, 1 mM Benzamidine) using a Sephadex G-25 column, before 3 h incubation with glutathione sepharose 4B (GE Life Sciences). AKAP79-D/D was eluted by overnight incubation with PreScission protease (GE Life Sciences). In the final purification step, AKAP79-D/D complex eluted between 0.4 and 0.5 column volumes from a Superdex-200 gel filtration column in 20 mM Hepes pH 7.5 and 200 mM NaCl (Supplementary Fig. 1b). AKAP79- Δ79–86 and W79A variants, in complex with the RIIα D/D domain, were expressed and purified in the same way. Gel filtration elution profiles confirm that deletion of amino acids 79–86 does not affect the peak elution volume of AKAP79- RIIα D/D upon gel filtration (Supplementary Fig. 1b). GST fusion proteins (Supplementary Fig. 1d, e, f), and calcineurin (WT and I396A/I400A; Supplementary Fig. 1h, i) were purified, using affinity to glutathione sepharose and size exclusion chromatography (SEC)[25], after expression in BL21 (DE3) *E. coli*.

For crystallography, human CaM was cloned into pET28-a for expression of untagged protein in *E. coli* BL21 (DE3) cells. CaM was expressed by overnight incubation at 37 °C in 3 L auto-inducing media[40], yielding a cell pellet of ~30 g. The pellet was resuspended in 150 mL lysis buffer (25 mM Tris pH 7.4, 500 mM NaCl, 1 mM EDTA, 2 mM DTT, 1 mM Benzamidine, 0.1 mg/mL$^{-1}$ lysozyme), and sonicated prior to high-speed centrifugation ($20,000 \times g$, 30 min). The clarified lysate was supplemented up to 5 mM $CaCl_2$, before incubation with phenyl sepharose (GE Life Sciences) for 3 h. After washing (50 mM Tris pH 7.4, 1 mM $CaCl_2$, 500 mM NaCl), CaM was eluted in 10 mL EDTA buffer (50 mM Tris pH 7.4, 1 mM EDTA). The eluate was diluted in buffer containing 20 mM Tris pH 8.5, 2 mM DTT, 1 mM EDTA to a final NaCl concentration of ~20 mM, enabling binding to a 1 mL Resource Q column (GE Life Sciences). CaM was eluted from the Resource Q column with a salt/pH gradient by varying the proportions of a mixture of buffers A (20 mM Tris pH 8.5, 20 mM NaCl, 1 mM benzamidine, 2 mM DTT, 1 mM EDTA) and B (20 mM Tris pH 7.2, 1 M NaCl, 1 mM benzamidine, 2 mM DTT, 1 mM EDTA). Finally, CaM was dialysed into water and lyophilized using a vacuum concentrator. This protocol yields ~1.5 mg purified CaM per L culture (Supplementary Fig. 1c). Vectors for expressing CaM12 (D20A/D56A) and CaM34 (D93A/D129A) variants were generated using site-directed mutagenesis of pET28-a-CaM, and these variants were expressed in the same way and purified using phenyl sepharose and gel filtration. For XL-MS experiments, CaM was expressed with a PreScission-cleavable GST tag, and purified by affinity to glutathione sepharose, PreScission cleavage and SEC[25]. Coomassie stained SDS-PAGE gels show the purity of biotinylated CaM, CaM, CaM12, and CaM34 (Supplementary Fig. 1g, j).

**Cross-linking coupled to mass spectrometry.** AKAP79-D/D and CaM were mixed at a 1:1 molar ratio to a final concentration of 1 mg/mL in 100 μL buffer containing 20 mM Hepes pH 7.5, 200 mM NaCl and 100 μM $CaCl_2$. Cross-linking was initiated by adding 2 μL H12/D12 DSS (Creative Molecules) from a 25 mM stock prepared in DMSO. After 30 min incubation at 30 °C in a thermomixer set to 300 rpm, reactions were terminated by addition of 5 μL from a 1 M stock of ammonium bicarbonate. Each 100 μg cross-linked sample was enzymatically digested with trypsin and subsequently enriched by SEC for cross-linked peptides. LC-MS/MS analysis was carried out on an Orbitrap Elite mass spectrometer (Thermo Scientific). The data were searched using xQuest[41] in ion-tag mode with a precursor mass tolerance of 10 p.p.m. For matching of fragment ions, tolerances of 0.2 Da for common-ions and 0.3 Da for cross-link ions were applied. Fragment ions were analyzed in an ion trap, trading lower mass accuracy for increased sensitivity and measuring speed. Potential cross-links were statistically validated from fragment ion spectra and false discovery rates (FDR) were assigned using xProphet[42]. Cross-linked AKAP79-D/D-CaM samples were prepared in biological triplicate, and each of these was measured as technical duplicates. For each experiment, the final data set was restricted to high-confidence unique cross-links with both ID scores >20 and FDRs <0.05. Spectra of potential crosslinks were also visually inspected to ensure good matches of ion series on both cross-linked peptide chains for the most abundant peaks.

**Pull-down assays**. For pull-down assays involving N-terminally FLAG-tagged AKAP79 variants, HEK293T cells (American Type Culture Collection) were initially cultured in DMEM supplemented with GlutaMAX, Penicillin/Streptomycin, and 10% (v/v) fetal bovine serum (all supplied by Gibco). 10 cm diameter plates were transfected at 60 % confluence using 4 μg pcDNA3.1-FLAG-AKAP79 and 16 μL Fugene 6 (Promega). Cells were transfected with WT, Δ33–48, Δ79–86, or Δ391–400 ('ΔPKA') AKAP79. 3 days after transfection, cells were collected and lysed in 1 mL HEK293T buffer (30 mM Tris pH 7.4, 150 mM NaCl, 1% Igepal CA-630, 1 mM EDTA, 1 mM benzamidine, 2 mM DTT). Following sonication, lysates were clarified by centrifugation, then split equally for incubation with either 20 μL CaM sepharose 4B (GE Healthcare) or 20 μL 8-AEA-cAMP-agarose (Biolog) beads. After overnight incubation, beads were washed with 6 × 1 mL HEK293T buffer, and finally protein was eluted with 40 μL 1 × LDS sample buffer (ThermoFisher). For CaM sepharose pull-downs, the clarified lysate and wash buffer were supplemented with 3 mM $CaCl_2$. FLAG-AKAP79 variants were detected using rabbit anti-FLAG primary antibody (Sigma), and goat anti-rabbit-HRP-conjugate secondary antibody (Bio-Rad). An additional experiment was performed to confirm that binding of WT AKAP79 to CaM-sepharose is $Ca^{2+}$-dependent. In this case, HEK293T cells were transfected with pcDNA3.1-AKAP79-V5 (WT or Δ79–86). The lysate from each plate was clarified by centrifugation, divided into two, and one half of the material was supplemented with 3 mM $CaCl_2$. Each half was incubated with 20 μL CaM sepharose 4B overnight, then washed with 6 × 1 mL HEK293T buffer (supplemented with 3 mM $CaCl_2$, where appropriate) the following morning, and eluted in 1 × LDS sample buffer. AKAP79-V5 was detected using anti-V5-HRP-conjugate antibody (ThermoFisher). For pull-down experiments using AKAP79-D/D complexes purified from bacteria, pull-down was performed using a similar procedure. In this case, 0.5 μg purified protein was added in each pull-down. Bound protein was detecting by first eluting with $Ca^{2+}$-free HEK293T buffer containing 1 mM EDTA, and then immunoblotting with anti-AKAP79 (Millipore) antibody.

To determine which CaM-binding site enables $Ca^{2+}$-dependent interaction between the N-terminus of AKAP79 and calcineurin, 1 μg GST-AKAP79 1–153 (WT or Δ79-86) was incubated with 2 μg calcineurin (WT or I396A/I400A) in pull-down buffer (50 mM Tris pH 7.5, 150 mM NaCl, 1% Igepal CA-630, 0.25% sodium deoxycholate, 1 mM EDTA, 1 mM benzamidine) containing 100 nM CaM. The samples were supplemented with 3 mM $CaCl_2$ where appropriate. The samples were incubated with 2 μL magnetic glutathione bead slurry (Pierce) for 4 h, washed 4 times in pull-down buffer supplemented with 50 nM CaM, and 3 mM $CaCl_2$ where appropriate. Finally, proteins were eluted from the magnetic beads in 50 μL 1 × LDS sample buffer. Calcineurin pull-down was detected using mouse anti-calcineurin antibody (Catalog number 610259, BD Biosciences, 1:500 dilution).

**Amplified luminescent proximity homogeneous assay screens**. To prepare biotin-labeled CaM for use in alphascreen assays, 1 mg purified CaM was incubated with a two-fold molar excess of EZ link NHS Biotin (Pierce) in 20 mM Na Hepes pH 7.5, 200 mM NaCl. Labeling was terminated after 30 min at 22 °C by supplementing with Tris-HCl pH 7.4 to a final concentration of 50 mM. Biotininylated CaM was separated from free biotin by SEC yielding >200 μL biotininylated CaM at 80 μM. In general, assays were performed in 50 μL buffer consisting of 20 mM HEPES pH 7.2, 100 mM NaCl, 0.1% BSA, 1 mM EGTA, 5 mM $MgCl_2$, 1 mM $CaCl_2$. $CaCl_2$ was excluded where appropriate. A PEPscreen was ordered from Sigma containing the following peptides (~2 mg each): 20-mer peptides walking through the N-terminus of AKAP79 in 5-amino-acid increments, starting at 21–40 and ending at 116–135 for a total of 20 peptides; and 20-mer peptides corresponding to AKAP79 (77–96) containing either WT sequence or the substitutions W79A, L82A, L85A, V86A, L82P. The following high (>95%) purity peptides were ordered from Biomatik: '9-mer' (AKAP79 positions 79–87), '11-mer' (77–87), '16-mer' (77–92), '20-mer' (77–96), and '26-mer' (77–102) both with and without the substitution L101A.

For all alphascreen experiments, GST-AKAP79 (1–153) and biotinylated CaM were first preincubated for 1 h at 22 °C in the presence of the appropriate peptide of unbiotinylated CaM additives. Biotin-CaM was included at 10 nM in all mixtures, whereas GST-AKAP79 (1–153) varied between experiments as follows: 3 nM (Fig. 2d, Fig. 3c–e, Supplementary Fig. 2c), 6 nM (Fig. 5f), and 8 nM (Fig. 2d). To determine the $K_{Ca}$ for CaM binding to calcineurin and AKAP79, the $CaCl_2$ concentration was varied to produce different free $Ca^{2+}$ concentrations on the basis of calculations using maxchelator[43]. After initial incubations, anti-GST acceptor beads (PerkinElmer) were added at 10 ng/μL. After 30 min, streptavidin donor beads were also added at 10 ng/μL. After 1 h further incubation with vigorous shaking in a thermomixer (1200 r.p.m.), light emission at 568 nm was measured following excitation with a 680 nm laser using a Mithras LB 940 (Berthold Technologies).

**Protein crystallization**. Peptide corresponding to AKAP79 positions 77–92 (GAWASLKRLVTRRKRS) was synthesized at 96% purity (Biomatik). Lyophilized CaM was resuspended in 10 mM $CaCl_2$ at 7.5 mg/mL and mixed with a 3:1 molar excess of peptide. Crystals were grown at 22 °C using hanging-drop vapor diffusion by mixing 1 μL of CaM solution with 1 μL mother liquor (2.4 M ammonium sulphate, 50 mM citrate pH 5.4, 0.3 M NDSB-195). After growth for one month, crystals were flash-frozen in liquid $N_2$ using mother liquor supplemented with 3.5 M ammonium sulphate.

---

**Table 1 X-ray diffraction data collection and refinement statistics**

| Data collection | |
|---|---|
| Beamline | DLS I24 |
| Space group | P 42 21 2 |
| Cell dimensions | |
| a, b, c (Å) | 76.46, 76.46, 128.99 |
| α, β, γ (deg) | 90, 90, 90 |
| Wavelength (Å) | 0.9685 |
| Resolution (Å) | 65.77-1.7 (1.73-1.7) |
| Rpim | 0.063 (0.354) |
| Half-set correlation $CC_{1/2}$ (%) | 0.990 (0.552) |
| I/σI | 10.1 (3.1) |
| Completeness (%) | 99.1 (99.6) |
| Multiplicity | 7.4 (7.5) |
| | |
| Refinement | |
| Resolution (Å) | 54.07 – 1.70 |
| No. of reflections | 78,207 |
| $R_{work}/R_{free}$ | 0.165, 0.195 |
| No. of atoms | 4704 |
| Average B factors | |
| Protein | 14.56 |
| Water | 26.66 |
| $Ca^{2+}$ | 6.78 |
| R.M.S. deviations | |
| Bond lengths | 0.009 |
| Bond angles | 0.904 |
| Ramachandran favoured/outliers (%) | 98.7/0 |

---

**X-ray diffraction data collection and processing**. The diffraction data were collected at the Diamond Light Source (Harwell, UK) beamline I24. Diffraction images were processed using iMosflm[44] and merging/scaling of reflection intensities was performed by AIMLESS[45]. AIMLESS indicated an appropriate resolution limit of 1.7 Å (1.73–1.7 Å shell: I/σI = 3.1, $CC_{1/2}$ = 0.552). Initial phases were obtained by molecular replacement in PHASER[46] using the C-lobe of CaM (residues 85–148, PDB 1IWQ) as the search model. This confirmed that two copies of the CaM–AKAP79 peptide complex are present in the asymmetric unit. Automated model building was performed in Buccaneer[31]. The model was refined by rounds of manual model building in Coot[32] and automated refinement in PHENIX[33]. Refinement was performed using 20 translation/libration/screw groups defined automatically by PHENIX. The structure was validated with MolProbity[47]. The data collection and refinement statistics are shown in Table 1. Stereo images of the $2F_o–F_c$ electron density map clipped to the AKAP79 peptide are shown in Supplementary Fig. 4b.

**Structure and sequence analysis**. Structural representations of the complex (chains B and D of the asymmetric unit unless otherwise stated) were rendered using Pymol (www.pymol.org). Secondary structure prediction was performed using Jpred3[48]. Alignments, and LOGO plots of AKAP5 sequences were retrieved from the ensembl database (www.ensembl.org). Distances were calculated using NCONT, and solvent accessibility was determined using AREAIMOL[49]. Automated H-bond identification was performed using UCSF Chimera[50], 3D structural homologs were identified using the DALI server[51], and pairwise structural alignments were performed using GESAMT[49].

**A-kinase activity reporter assays**. Fluorescence-based recordings of PKA activity were performed in HEK293T cells attached to 96-well plates. HEK293T cells were cultured and transfected in the same way as for pull-down assays. In this case, cells were transfected at 60 % confluence in 6-well plates using 1 μg DNA/4 μL Fugene 6 per well. The DNA consisted of a mixture of 0.3 μg pcDNA3.1-FLAG-AKAP79 (WT or Δ79-86) and 0.7 μg AKAR4 DNA. 2 days after transfection, the cells were detached using trypsin and resuspended in 3 mL supplemented DMEM, then re-plated at 100 μL/well in 96-well black-walled plates (Molecular Probes) that had been coated overnight with poly-l-lysine (Sigma). The following morning, the media was exchanged for PBS (100 μL per well), and FRET ratios were immediately determined using a Mithras LB 940 plate-reader (Berthold Technologies) controlled by a computer running Mikrowin 2000 software. To enable recordings using AKAR4[30], the plate reader was fitted with a 420 nm excitation filter (Berthold no. 39452), and emission filters at 485 nm (no. 40271) and 535 nm (no. 40273).

**Protein phosphatase assays**. Protein phosphatase activity was determined by measuring phosphate release from phosphorylated RII peptide

(DLDVPIPGRFDRRVpSVAAE) synthesized at >95% purity (Biomatik). Assays were performed with either WT or I396A/I400A calcineurin at 20 nM in 50 µL reactions containing different concentrations of CaM in phosphatase assay buffer (20 mM Na Hepes, 100 mM KCl, 0.1% BSA, 1 mM EGTA, 6 mM MgCl₂, 1 mM CaCl₂). Each assay was initiated by adding pRII substrate (75 µM final concentration), and terminated after 1 h at 30 °C by addition of 50 µL Biomol Green (Enzo Life Sciences). Phosphate release was determined by measuring absorbance at 630 nm using an ELx800 spectrophotometer (Biotek) after color had been allowed to develop for 30 min at 22 °C.

**Motif-based scanning for 1-4-7-8 CaM-binding sites**. Scansite 3 was used to perform searches for putative 1-4-7-8 class CaM-binding motifs[52]. To identify potential 1-4-7-8 motifs, we assembled a position-specific scoring matrix file guided by contacts between AKAP79 and CaM in the crystal structure: the strongest weighting was assigned to the first anchor tryptophan, followed by hydrophobic amino acids at positions 4, 7, and 8. Conservative substitutions that could be accommodated according to the crystal structure were assigned intermediate weightings. The basic amino acids lysine and arginine were equally weighted at positions '5' and '6'. The full matrix is shown in supplementary Fig. 5. The matrix was used to search *Homo sapiens* protein sequences within the SwissProt database (20252 total input proteins). 570 predicted sites were found, with a median score = 0.633, and median absolute deviation = 0.108. The top 100 highest scoring predicted sites are listed in supplementary Table 3.

**Statistical analysis**. Densitometry was performed using ImageJ[53]. Two-tailed student's *t* tests were used to calculate *p* values for the differences between mean values (*n* ≥ 3 for all experiments). Densitometry was performed using ImageJ[53]. Curve fitting to determine $K_{Ca}$ and $K_i$ values was performed by fitting data to Hill functions by iterative least squares minimization using ORIGIN software (OriginLab). $K_{CaM}$ co-efficients for calcineurin activation were calculated by fitting to Michaelis-Menten functions containing a cooperativity variable (Supplementary Fig. 3).

**Data availability**. The coordinate and structure factor data of the CaM—AKAP79 (77–92) complex has been deposited in the Protein Data Bank (PDB) with the accession code 5NIN. Uncropped images of blots and gels are shown in Supplementary Fig. 6. The other data are available from the corresponding author upon reasonable request.

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

## Acknowledgements

We thank Ambrose Cole, Duncan Laverty and staff at beamline I24 at the Diamond Light Source for help with data collection; Kanchan Chargar for assistance with cell culture; and Stuart Martin for help calibrating the plate reader for alphascreen and FRET recordings. MGG is a Wellcome Trust and Royal Society Sir Henry Dale fellow (104194/Z/14/Z), and receives support from the BBSRC (BB/N015274/1). F.S. is funded by the German Science Foundation Emmy Noether Programme (STE 2517/1-1) and is grateful for support from the DFG Collaborative Research Center (SFB) 969 and the Wellcome Trust (Grant 095951). R.A. was supported by ERC AdG-670821 (Proteomics 4D).

## Author contributions

M.G.G. directed the project, performed alphascreen, phosphatase and pull-down assays, collected the X-ray diffraction data, and solved and analyzed the structure. N.P.: Prepared purified proteins, performed pull-down assays, and crystallized the complex. F.S.: Performed mass spectrometry experiments and processed the XL-MS data. R.A.: Supervised XL-MS experiments. M.G.G.: Prepared the manuscript with input from all the authors.

## Additional information

**Competing interests:** The authors declare no competing financial interests.

