## [Peer Review file · Nature Communications]

Reviewers' comments:

Reviewer #1 (Remarks to the Author):

The paper is about the regulation of PKA kinase activity by AKAP79 interacting with calcineurin in a Ca²⁺/CaM-dependent manner. First, the sequence coverage of the AKAP79/CaM interface has been characterized by mass spec-based cross-linking. The most frequent interaction was found between Lys99(AKAP79) and Lys94(CaM). Next, by using an AKAP79 deletion mutant (Delta79-86) it has been shown that CaM binding is impaired. By combining these findings with a bioinformatics analysis it has been suggested that CaM is bound by a novel 1-4-7-8 binding motif. Further, it has been shown that when using the AKAP79 deletion mutant (Delta79-86) calcineurin binding – by using the KM of calcineurin phosphatase activity assay – is considerably reduced and PKA activity is increased. The remaining results are basically about a crystal structure of CaM in the presence an AKAP79 peptide covering residues 77-92. Overall the findings are predominantly been put together aiming to demonstrate that a novel CaM-target binding motif has been discovered. In my view however this claim, based on the data presented, remains uncertain for the reasons outlined below:

- a) The initial basis of the claim is that CaM binding is impaired in the AKAP79 deletion mutant (Delta79-86). However, this sequence segment is outside of the main interaction sites found by cross-linking (Figure 1g). Neither wt-AKAP79 nor the Delta79-86 variant has been structurally or at least biophysically characterized by any means, leaving various alternative options open why this mutant is impaired.
- b) The bioinformatics analysis presented in Figure 2a appears to be far-fetched and/or is insufficiently described in the Methods section. There are much more than six CaM/"target" complexes in the PDB. What have been the selection criteria? I can't follow how the limited description of this analysis has been used to justify the focus on residues N-terminal to the sites found by cross-linking. A more experimental approach, such e.g. a scanning peptides near and within the found AKAP79/CaM binding would have been more convincing.
- c) A crystal structure of an AKAP79 peptide 77-92 and CaM is described in quite length, leading to several novelty claims about the peptide conformation, the CaM conformation and Ca²⁺ occupancy. However, it has also become obvious from several recent publications that there is a significant likelihood that peptide complexes are in conflict with structures of protein targets of the same sequences. No effort has been made to validate those structural data experimentally, to reconcile the data with the cross-linking data shown in Figure 1g (which I trust most, as they address the interactions in the context of full length proteins). Assuming that a structure with AKAP70 protein is not feasible, one would minimally expect a validation analysis of full-length AKAP79 binding to CaM, using single residue mutations both addressing the main sites found by cross-linking and in the structure, with the aim of reconciling both data sets.
- d) Where there are experimental discrepancies, e.g. partial Ca²⁺ occupancy versus the observed high cooperativity, one would have expected some sort of experimental clarification, instead we find wordy explanations. An obvious experimental approach would be on testing Ca²⁺ binding site CaM mutants.
- e) The claim of generalization for the apparently novel 1-4-7-8 motif in the second part of the discussion section is without any real substance, in the absence of an attempt of experimental verification. An honest discussion about the present shortcomings and limitations of the presented analysis in the Discussion section would have been more appreciated.

Further points:

The paper is full of largely unjustified superiority statements. I strongly advice to tone those down,

you are not going to win anything by those.

AKAP79 has been purified in the presence of PKA RII- α , as stated in the Methods Section. Please provide a reference for providing insight into the logic behind. Can the authors rule out that binding of PKA RII- α has impact on CaM and calcineurin binding, including the cross-linking experiments?

In the alphascreen assay, an AKAP70 peptide covering residues 77-96 disrupts binding of wt-AKAP79 to CaM. To analyze these data, authors calculate a K_i value, which is not defined in the manuscript. Is it estimated from a simple hill/hyperbolic equation? If true it does not make sense to compare this number with K_D calculated elsewhere. To properly determine affinity constants the data need to be fitted according to a competition model where concentrations of all species are taken into account.

The section on AKAP79-calcineurin binding is confusing as the Michaelis constant from turn over kinetics is used as a measure of binding. Please show all relevant kinetic values, possibly in the Supplement, to allow proper judgment. Is there no option to measure binding directly?

Regarding the crystal structure, there is no statement about PDB deposition. Isn't this required? Table 1 contains several beginner mistakes, such as missing dimensions and unreasonable precision values. Please consult an experienced structural biologist if uncertain. Parts of Supplementary Figure 2 are of poor quality, namely panels a and c (e.g. side chain of Phe12 literally not visible).

Figure 6c seems to imply that the CaM binding motif in AKAP79 is found in other proteins with the same CaM binding function as well. This has not been experimentally validated, so speculation should not be mixed with experimental evidence shown in panels a and b.

Reviewer #2 (Remarks to the Author):

Comments to the Authors:

In this manuscript, the authors describe interaction studies between AKAP79 and calmodulin (CaM) that they consider to be mediated via a novel 1-4-7-8 CaM binding motif. The AKAP79/CaM interaction is studied by chemical cross-linking in combination with mass spectrometry where a large number of cross-links were observed between Lys-94 in CaM and Lys-99 in AKAP79. Removal of the amino acid stretch 79-86 (WASLKRLV) in AKAP79 abolished the interaction with CaM. Additionally, the authors performed X-ray crystallography of CaM with AKAP79 amino acids 77-92 (GAWASLKRLVTRRKRS).

I performed a prediction analysis of CaM-binding motifs in the AKAP79 sequence revealing two potential CaM-binding sites (shown below). Interestingly, the predicted CaM binding motif (aa 43-58) was also detected in the cross-linking analysis (Figure 1g), confirming this additional CaM binding site in AKAP79.

From their cross-linking experiments, the authors conclude the existence of a novel type of CaM-binding motif in AKAP79. However, I am not convinced that the CaM binding motif found for AKAP79 is really a 1-4-7-8 motif. The sequence stretch aa 79-92 in AKAP79, which was shown to be responsible for mediating AKAP79/CaM interaction, contains two hydrophobic anchor amino acids: WASLKRLVTRRKRS (hydrophobic anchor amino acids are underlined). In fact, many CaM-binding motifs have been described in the literature showing a variable spacing of the hydrophobic anchor amino acids. The authors exclusively mention the existence of 1-5-10 and 1-8-14 CaM-binding motifs, which is a too simplified view. I recommend checking existing literature on the variety of CaM-binding motifs. For example, there is a 1-5-8-26 CaM-binding motif existing in Munc13 proteins (see publication attached to this review). The sequence of AKAP79 contains Leu-101 (corresponding to position 23 of the CaM-binding motif) that might serve as additional anchor residue (WASLKRLVTRRKRSSESSKQKPL), similar to the CaM-binding motif in Munc13. As such, the list of proteins exhibiting potential 1-4-7-8 CaM-motifs (Suppl. Table 3) is not useful as it does not give information on hydrophobic anchor residues that are located more C-terminal compared to the classical CaM-binding motifs.

Almost equally important as the hydrophobic anchor residues is the presence of a cluster of basic amino acids. When checking the binding sequence in AKAP79 it is immediately apparent that the CaM-binding motif in AKAP79 contains this cluster of basic amino acids: WASLKRLVTRRKRS (cluster of basic amino acids is underlined).

The authors performed cocrystallization experiments between CaM and the AKAP79 amino acids 77-92 comprising the 1-8-14 motif. It is a known fact that short CaM-target peptides do not always accurately reflect the binding behavior of longer peptides or the full length protein. One example is the abovementioned Munc13 protein where short peptides reflected a more compact formation of the CaM/Munc13 peptide complexes. But when studying longer peptides, it became apparent that the structure of the CaM/Munc13 complex is only correctly reflected if the additional hydrophobic anchor amino acid at position 26 is present (see publication attached). It is therefore crucial to use C-terminally elongated peptides containing all potential hydrophobic anchor residues. A CaM-binding motif in AKAP79 with an additional, more C-terminal, anchor residue would also be in agreement with the finding that CaM-binding is abolished when amino acid 78-86 are removed in AKAP79. Also, the additional hinge point at Leu-101 in AKAP79 would be in agreement with the "hot-spot" cross-link between Lys-94 in CaM and Lys-99 in AKAP79.

I therefore suggest that the authors perform additional X-ray crystallography experiments with C-terminally elongated AKAP79 peptides, comprising Leu-101.

Additional comment on MS analysis: The authors should aim for recording MS/MS data with high resolution and high mass accuracy. In this work, mass tolerance was set to 0.3 Da (page 23) for cross-linked peptides, which will not allow an unambiguous identification of cross-linked products, especially for highly charged species.

Prediction of CaM-binding motifs in AKAP79:

calcium.uhnres.utoronto.ca

UniProtKB - P24588 (AKAP5_HUMAN)

```
....1 METTISEIHV ENKDEKRSAE GSPGAERQKE KASMLCFKRR KKAALKKPK
..... 0000000000 0000000000 0000000135 7799999999 9999999753

...51 AGSEAADVAR KCPQEAGASD QPEPTRGAWA SLKRLVTRRK RSESSKQKPK
..... 1100000000 0000000000 0000001111 1111111111 1111110000

..101 LEGEMQPAIN AEDADLSKKK AKSRLKIPCI KFPRGPKRSN HSKIIEDSDC
..... 0000000000 0000000000 0000000000 0000000000 0000000000

..151 SIKVQEEAEI LDIQTQTPLN DQATKAKSTQ DLSEGISRKD GDEVCESNVS
..... 0000000000 0000000000 0000000000 0000000000 0000000000

..201 NSTTSGEKVI SVELGLDNGH SAIQTGTLIL EEIETIKEKQ DVQPQQASPL
..... 0000000000 0000000000 0000000000 0000000000 0000000000

..251 ETSETDHQQP VLSDVPPLPA IPDQQIVEEA SNSTLESAPN GKDYESTEIV
..... 0000000000 0000000000 0000000000 0000000000 0000000000

..301 AEETKPKDTE LSQESDFKEN GITEEKSKSE ESKRMEPIAI IITDTEISEF
..... 0000000000 0000000000 0000000000 0000000000 0000000000

..351 DVTKSKNVPK QFLISAENEQ VGVFANDNGF EDRTSEQYET LLIETASSLV
..... 0000000000 0000000000 0000000000 0000000000 0000000000

..401 KNAIQLSIEQ LVNEMASDDN KINLLQ
..... 0000000000 0000000000 0000000
```

Reviewer #3 (Remarks to the Author):

This manuscript details the very interesting finding of a novel, previously unseen, mode of calmodulin (CaM) - substrate binding. Given the importance of CaM in sensing intracellular changes in calcium concentration and using this to modulate the activities of a great many substrate proteins, this finding should prove of great interest. The manuscript is well-written and the work presented appears to be well done. I do have a couple of minor issues I would like to see addressed.

1) The authors have measured CaM binding to calcineurin to be 30 ± 12 nM. This is at odds with published data that is in the low picomolar range (e.g. O'Donnell et al. (2011) *Proteins* 79:765 and Quintana et al. (2005) *BBRC* 334:674). The authors need to address this discrepancy.

2) Why was the alpha-screen used to measure binding? There are multiple methods available that would not require coupling of the proteins to large beads, or fusion to other proteins, that could interfere with the measurements. Were the alpha-screen results validated against another method?

3) Figure 3a shows the locations of the isoleucine to alanine mutations made in the calcineurin CaM binding region. Although the figure is adequate to show the locations of the mutations, the full sequence of the CaM binding region is not shown, which I find a little misleading.

Reviewers' comments:

Reviewer #1 (Remarks to the Author):

The paper is about the regulation of PKA kinase activity by AKAP79 interacting with calcineurin in a Ca²⁺/CaM-dependent manner. First, the sequence coverage of the AKAP79/CaM interface has been characterized by mass spec-based cross-linking. The most frequent interaction was found between Lys99(AKAP79) and Lys94(CaM). Next, by using an AKAP79 deletion mutant (Delta79-86) it has been shown that CaM binding is impaired. By combining these findings with a bioinformatics analysis it has been suggested that CaM is bound by a novel 1-4-7-8 binding motif. Further, it has been shown that when using the AKAP79 deletion mutant (Delta79-86) calcineurin binding – by using the KM of calcineurin phosphatase activity assay – is considerably reduced and PKA activity is increased. The remaining results are basically about a crystal structure of CaM in the presence an AKAP79 peptide covering residues 77-92. Overall the findings are predominantly been put together aiming to demonstrate that a novel CaM-target binding motif has been discovered. In my view however this claim, based on the data presented, remains uncertain for the reasons outlined below:

a) The initial basis of the claim is that CaM binding is impaired in the AKAP79 deletion mutant (Delta79-86). However, this sequence segment is outside of the main interaction sites found by cross-linking (Figure 1g). Neither wt-AKAP79 nor the Delta79-86 variant has been structurally or at least biophysically characterized by any means, leaving various alternative options open why this mutant is impaired.

Response 1: The reviewer points out that the binding sequence W79-V86 in AKAP79 does not exactly overlap with the most densely interlinked lysines K90, K96, and K99. As a lysine-reactive crosslinker was used in this study, one could only expect K83 within the interaction helix to be directly detected. Protein interfaces that become buried/occluded upon binding of an interaction partner are often found to be excluded or at least less prevalent in crosslinking studies. Presumably in this case tight CaM-AKAP79 contacts impede crosslinker access to K83. Besides K83, lysines 90, 96, and 99 are the three closest lysines to the helical motif (the uneven distribution of lysines in AKAP79 is shown in **Fig. 1g**). The clear dominance of links to these lysines is therefore consistent with the other experiments in the manuscript.

Regarding evidence that AKAP79 amino acids 79-86 are the key site for interacting with CaM, our revised manuscript includes several additional experiments that support this conclusion. These include a modified HEK cell lysate pull-down experiment (**Fig. 3a**), which includes control binding to cAMP agarose. AKAP79 associates with cAMP agarose through its ability to associate with type II PKA regulatory subunits therefore cAMP agarose binding shows that the anchoring protein is not impaired in binding abother interaction partner. We found that whereas wild-type full-length AKAP79 was recovered with both CaM sepharose and cAMP agarose, AKAP79 Δ 79-86 associated with cAMP agarose but not CaM sepharose (lane 3, **Fig. 3a**). This indicates that the Δ 79-86 variant is capable of binding to PKA RII subunits but not to CaM. A variant lacking the binding site for PKA (Δ 391-400) was recovered with CaM sepharose but not cAMP agarose, as expected (lane 4, **Fig. 3a**). We have also performed pull-down experiments with full-length AKAP79 purified from bacteria. The gel filtration elution profiles of wild-type and Δ 79-86 AKAP79, purified from bacteria in complex with RII α D/D, show that the two variants elute with similar peaks at \sim 0.43 column volumes (**Supplementary Fig. 1b**). Material eluting in the peak fractions was used as the input for pull-down experiments shown in **Fig. 5d**. We now also shown that the single point mutation W79A is sufficient to prevent pull-down of purified full-length AKAP79-RII α D/D using CaM sepharose (**Fig. 5d**, lanes 5 & 6).

b) The bioinformatics analysis presented in Figure 2a appears to be far-fetched and/or is insufficiently

described in the Methods section. There are much more than six CaM/"target" complexes in the PDB. What have been the selection criteria? I can't follow how the limited description of this analysis has been used to justify the focus on residues N-terminal to the sites found by cross-linking. A more experimental approach, such e.g. a scanning peptides near and within the found AKAP79/CaM binding would have been more convincing.

Response 2: We have extended the text under 'Structure and sequence analysis' to more clearly explain the rationale and method behind this analysis (p. 45, last seven lines; page 45, first 4 lines). Very briefly, the five structures are intended to be representative and correspond to two 1-5-10 motifs, two 1-8-14 motifs, and one atypical motif (Munc13). More importantly, we have moved this analysis to **Supplementary Figure 2a**, and replaced it with a scanning experiment as suggested by the reviewer (new **Fig. 2c & d**). We compared the ability of 20 different peptides, derived from moving in 5-amino-acid increments along the sequence of the AKAP79 N-terminus, to disrupt interaction between AKAP79 (1-153) and CaM. Peptides corresponding to positions 71-90, and 76-95 were the most potent inhibitors of the interaction (**Fig. 2d**, peptides K & L). These are the only two peptides that incorporate the full 79-86 helix. We acknowledge that this experiment is a much stronger link between the crosslinking and subsequent experiments, and thank the reviewer for his/her suggestion.

c) A crystal structure of an AKAP79 peptide 77-92 and CaM is described in quite length, leading to several novelty claims about the peptide conformation, the CaM conformation and Ca²⁺ occupancy. However, it has also become obvious from several recent publications that there is a significant likelihood that peptide complexes are in conflict with structures of protein targets of the same sequences. No effort has been made to validate those structural data experimentally, to reconcile the data with the cross-linking data shown in Figure 1g (which I trust most, as they address the interactions in the context of full length proteins). Assuming that a structure with AKAP70 protein is not feasible, one would minimally expect a validation analysis of full-length AKAP79 binding to CaM, using single residue mutations both addressing the main sites found by cross-linking and in the structure, with the aim of reconciling both data sets.

Response 3: Regarding single-residue mutations in AKAP79, we have performed additional experiments presented in **figure 5c & d** to corroborate the interaction mode observed in the crystal structure. We started by measuring how point mutations in peptides corresponding to AKAP79 positions 77-96 affect the ability of this peptide to prevent AKAP79 (1-153) – CaM association (**Fig. 5c**). The mutation W79A led to the most profound reduction in peptide potency. Crucially, we next assessed the effect of incorporating this point mutation in the context of full-length AKAP79. Consistent with W79 serving as the key anchoring position for CaM association (**Fig. 5b**), substitution of this residue with alanine prevented binding of full-length AKAP79 to CaM sepharose (**Fig. 5d**, lane 6).

d) Where there are experimental discrepancies, e.g. partial Ca²⁺ occupancy versus the observed high cooperativity, one would have expected some sort of experimental clarification, instead we find wordy explanations. An obvious experimental approach would be on testing Ca²⁺ binding site CaM mutants.

Response 4: The revised manuscript includes an additional experiment in which we compared the ability of either wild-type CaM, or Ca²⁺-binding deficient mutants of either the N-lobe (CaM12) or C-lobe (CaM34) to inhibit interaction between biotin-CaM and AKAP79 (1-153). We found that CaM12 inhibited the interaction with a similar K_i (91±6 nM, red line, **Fig. 6e**) to wild-type CaM (108±8 nM, black line), consistent with the notion that Ca²⁺ binding to the CaM N-lobe is not required for interaction with AKAP79. CaM34 (K_i = 930±130 nM, blue line, **Fig. 6e**) was a significantly less potent inhibitor of the interaction, which supports the idea that generation of the open conformation of the C-lobe by Ca²⁺ binding enables interaction with AKAP79.

e) The claim of generalization for the apparently novel 1-4-7-8 motif in the second part of the discussion section is without any real substance, in the absence of an attempt of experimental verification. An honest discussion about the present shortcomings and limitations of the presented analysis in the Discussion

section would have been more appreciated.

Response 5: We have modified text regarding generalization of the 1-4-7-8 motif to highlight the limitations of our analysis (p. 21, lines 19-21) and emphasize that it will be important to experimentally verify putative 1-4-7-8 motifs.

Further points:

The paper is full of largely unjustified superiority statements. I strongly advise to tone those down, you are not going to win anything by those.

Response 6: We have taken this point on board, and factored it into our re-drafting of the manuscript.

AKAP79 has been purified in the presence of PKA RII-alpha, as stated in the Methods Section. Please provide a reference for providing insight into the logic behind. Can the authors rule out that binding of PKA RII-alpha has impact on CaM and calcineurin binding, including the cross-linking experiments?

Response 7: We find that AKAP79 purified in the absence of the dimerization and docking (D/D) domain of RII subunits forms soluble aggregates that elute in the void volume upon gel filtration. The RII binding site within AKAP79 consists of an amphipathic helix spanning amino acids 391-408 that docks to a shallow hydrophobic groove presented by an X-type four helix bundle formed by the dimerization of the first 45 amino acids of RII (relevant structural references include Kinderman et al., Mol Cell, 2006, PMID 17081990; Gold et al., Mol Cell, 2006, PMID 17081989; reviewed by Taylor et al., Nature reviews molecular cell biology, 2012, PMID 22992589). Presumably, when the hydrophobic face of the anchoring helix is not occupied it can trigger aggregation. Since the RII anchoring site is very close to the C-terminus of the anchoring protein, we do not anticipate that the presence of the RII D/D domain alters interactions between calcineurin or CaM within the N-terminus of the anchoring protein.

In the alphascreen assay, an AKAP70 peptide covering residues 77-96 disrupts binding of wt-AKAP79 to CaM. To analyze these data, authors calculate a K_i value, which is not defined in the manuscript. Is it estimated from a simple hill/hyperbolic equation? If true it does not make sense to compare this number with K_D calculated elsewhere. To properly determine affinity constants the data need to be fitted according to a competition model where concentrations of all species are taken into account.

Response 8: The reviewer is correct in his/her assumption that K_i values were determined throughout by fitting to a simple Hill equation. We now define K_i on p. 10, line 4. We agree that it is inappropriate to relate these values to the K_D value determined elsewhere, so have removed this comparison from the text.

The section on AKAP79-calcineurin binding is confusing as the Michaelis constant from turn over kinetics is used as a measure of binding. Please show all relevant kinetic values, possibly in the Supplement, to allow proper judgment. Is there no option to measure binding directly?

Response 9: The reviewer rightly points out that half-maximal CaM concentrations for binding and activating calcineurin are not necessarily the same. Indeed, the evidence is that CaM binding occurs at substantially lower concentrations than necessary to activate the enzyme. We have modified the text to state that the experiment presented in **figure 4c** suggests that calcineurin activation is not required for Ca^{2+} -dependent interaction with the N-terminus of AKAP79 but that it does not rule out the possibility that CaM binding is required (p. 12, lines 3-6). Although these experiments were not performed with the aim of accurately determining values for k_{cat}/V_{max} , WT and mutant assays were performed with the same substrate so our relative values for V_{max} can be compared. We have included all relevant kinetic data in a new supplementary figure (**Supplementary Fig. 3**). Our calculated K_{CaM} for activation of wild-type calcineurin is comparable to that determined by Quintana et al., Biochem Biophys Res Commun, 2005, PMID 16009337 (30 nM vs 15 nM determined in Figure 1 of Quintana et al, 2005). If we exclude our measurements at 5 μ M and 10 μ M CaM (3 μ M is the highest concentrations used by Quintana et al.), and

model in the same way, we calculate a $K_{CaM} = 16 \pm 3$ nM ie very close to the figure obtained in this other study. This curve fit is shown below.

Regarding the possibility of measuring CaM binding to calcineurin directly. It has proved to be challenging to accurately determine a definitive equilibrium constant for wild-type calcineurin (compare, e.g., Quintana et al., 2005 to Perrino et al., Eur J Biochem, 2002, PMID 12135494) so we decided against undertaking experiments in this direction.

Regarding the crystal structure, there is no statement about PDB deposition. Isn't this required? Table 1 contains several beginner mistakes, such as missing dimensions and unreasonable precision values. Please consult an experienced structural biologist if uncertain. Parts of Supplementary Figure 2 are of poor quality, namely panels a and c (e.g. side chain of Phe12 literally not visible).

Response 10: We have included a statement including the PDB accession code (5NIN) under 'Data availability'. The original version of Table 1 did include the cell lengths and angles in the two lines underneath 'Cell dimensions', with the table formatted in the same style as equivalent tables in other structural publications (e.g., Godoy et al., Nature Communications, 2017. PMID 28345596). We have modified the table in one way: we have removed the statistic 'Rmeas', which is not a good indicator of quality at high multiplicity as in this case. Our data quality indicators (including Rpim, CC1/2) suggest that we have chosen a suitable resolution cut-off (see Karplus & Diederichs, Linking crystallographic model and data quality, Science, 2012. PMID 22628654). Regarding precision values, please see below an output from POLYGON, which plots the refinement measures from our structure in relation to other structures of similar resolution (high frequency is shown by darker grey). Ranges are specified by numbers printed in red. This analysis suggests that our refinement was reasonable. We have also separately attached the PDB validation report for the structure.

We thank the reviewer for directing us to modify the supplementary structural figures. We have included additional labels in **supplementary figure 4a**, and the structure in **supplementary figure 4c** is now presented from a different angle so Phe12 is clearly visible.

Figure 6c seems to imply that the CaM binding motif in AKAP79 is found in other proteins with the same CaM binding function as well. This has not been experimentally validated, so speculation should not be mixed with experimental evidence shown in panels a and b.

Response 11: We have removed the word ‘Class’ from below AKAP79 in this figure (now **Fig. 7a**), and have also removed the sequences derived from Gravin and Cav2.1 from the sequence alignment (**Fig. 7d**).

We thank the reviewer for his/her many insightful and constructive comments.

Reviewer #2 (Remarks to the Author):

In this manuscript, the authors describe interaction studies between AKAP79 and calmodulin (CaM) that they consider to be mediated via a novel 1-4-7-8 CaM binding motif. The AKAP79/CaM interaction is studied by chemical cross-linking in combination with mass spectrometry where a large number of cross-links were observed between Lys-94 in CaM and Lys-99 in AKAP79. Removal of the amino acid stretch 79-86 (WASLKRLV) in AKAP79 abolished the interaction with CaM. Additionally, the authors performed X-ray crystallography of CaM with AKAP79 amino acids 77-92 (GAWASLKRLVTRRKRS).

I performed a prediction analysis of CaM-binding motifs in the AKAP79 sequence revealing two potential CaM binding sites (*shown below*). Interestingly, the predicted CaM binding motif (aa 43-58) was also detected in the cross-linking analysis (Figure 1g), confirming this additional CaM binding site in AKAP79.

UniProtKB - P24588 (AKAP5_HUMAN)

```

...1 METTISEIHV ENKDEKRSAE GSPGAERQKE KASMLCFKRR KKAALKPK
..... 0000000000 0000000000 0000000135 7799999999 999999753

...51 AGSEAADVAR KCPQEAGASD QPEPTRGAWA SLKRLVTRRK RSESSKQKPK
..... 1100000000 0000000000 0000001111 1111111111 1111110000

..101 LEGEMQPAIN AEDADLSKKK AKSRLKIPCI KFPRGPKRSN HSKIIEDSDC
..... 0000000000 0000000000 0000000000 0000000000 0000000000

..151 SIKVQEEAEI LDIQTQTPLN DQATKAKSTQ DLSEGISRKD GDEVCESNVS
..... 0000000000 0000000000 0000000000 0000000000 0000000000

```

Response 1: We thank the reviewer for bringing this prediction to our attention (the boundaries of the predicted site in the attachment provided by the reviewer are 33-48, as shown above). We note that this prediction was generated using a binding site search of the Calmodulin Target Database (Yap et al. Calmodulin Target Database, J Struct Funct Genomics, 2000). Our understanding from discussion with others in the field is that for a long time it was assumed that this site (33-48) was the CaM interaction site in AKAP79 but binding and deletion studies had repeatedly failed to confirm this. We agree, however, that it is important to exclude the possibility that CaM binds to this site in our study, and have performed two additional experiments to address this. Both experiments indicate that this region does not bind CaM. First, we have performed a peptide ‘walk’ through the N-terminus of AKAP79 to identify sequences that inhibit the interaction between CaM and AKAP79 (1-153). Peptides ‘C’ (31-50) and ‘D’ (36-55) did not inhibit the interaction - unlike peptides including amino acids 79-86 did (peptides K & L, **Fig. 2d**). Conclusively, we also performed CaM sepharose pull-down experiments using AKAP79 deletion mutants expressed in HEK cells. These experiments show that both wild-type and Δ 33-48 AKAP79 bind to CaM sepharose with comparable efficiency whereas the Δ 79-86 variant does not bind (**Fig. 3a**).

Regarding the identification of crosslinks in the vicinity of amino acids 33-48. We detected 7 interlinks in this region (involving AKAP79 lysines K41, K48, and K50) compared to 31 within the K91-K96-K99 cluster. Additionally, we recorded a third cluster of 5 interlinks involving AKAP79 positions K118-K120-K122. These three clusters in AKAP79 correspond to the three lysine-rich poly-basic regions of AKAP79 (these three regions were first noted by Dell’Acqua et al., EMBO, 1998, PMID 9545238). The much higher concentration of CaM interlinks to site B is consistent with CaM binding to amino acids 79-86. Finally, the failure of the Calmodulin Target Database search engine to assign a high score to the 79-86 sequence is to be expected since the search algorithm is based on homology to established CaM interaction motifs: the 79-86 sequence does not closely conform to any previously reported CaM target sequence.

From their cross-linking experiments, the authors conclude the existence of a novel type of CaM-binding motif in AKAP79. However, I am not convinced that the CaM binding motif found for AKAP79 is really a 1-4-7-8 motif. The sequence stretch aa 79-92 in AKAP79, which was shown to be responsible for mediating AKAP79/CaM interaction, contains two hydrophobic anchor amino acids: WASLKRLVTRRKRS (hydrophobic anchor amino acids are underlined). In fact, many CaM-binding motifs have been described in the literature showing a variable spacing of the hydrophobic anchor amino acids. The authors exclusively mention the existence of 1-5-10 and 1-8-14 CaM-binding motifs, which is a too simplified view. I recommend checking existing literature on the variety of CaM-binding motifs. For example, there is a 1-5-8-26 CaM-binding motif existing in Munc13 proteins (see publication attached to this review). The sequence of AKAP79 contains Leu-101 (corresponding to position 23 of the CaM-binding motif) that might serve as additional anchor residue (WASLKRLVTRRKRSSESSKQKPL), similar to the CaM-binding motif in Munc13. As such, the list of proteins exhibiting potential 1-4-7-8 CaM-motifs (Suppl. Table 3) is not useful as it does not give information on hydrophobic anchor residues that are located more C-terminal compared to the classical CaM-binding motifs.

Response 2: We thank the reviewer for attaching the interesting article concerning the Munc13 – CaM interaction. In the introduction of the original manuscript we do mention the existence of other motifs besides 1-5-10 and 1-8-14. The relevant passage is as follows: ‘*Outside of these predominant classes, in a few cases Ca²⁺/CaM binds in an extended conformation to longer recognition sequences (Tidow & Nissen, 2013 review). Prior to this study, in all cases where CaM binding depends on Ca²⁺, all four EF hands have been found to bind Ca²⁺ in high-resolution structures. In addition, with a single exception (Yamauchi et al., 2003), the terminal hydrophobic anchor positions are separated by no fewer than eight amino acids.*’ In the revised manuscript, we include additional text referring to the study by Rodriguez-Castandena et al. as an example of a longer recognition sequence on p. 3, lines 18-19, and we reference this paper again on p. 10, lines 17-19. We have also re-worded an earlier sentence to clarify that not all CaM targets fall within the 1-5-10 and 1-8-14 classes (p.3, lines 7-9). The possibility that additional hydrophobic elements are present further downstream of the ‘WASLKRLV’ sequence is addressed in ‘Response 4’ below. We have also extended putative CaM binding sequences listed in **supplementary table 3** out to 30 amino acids after the first hydrophobic anchor position.

Almost equally important as the hydrophobic anchor residues is the presence of a cluster of basic amino acids. When checking the binding sequence in AKAP79 it is immediately apparent that the CaM-binding motif in AKAP79 contains this cluster of basic amino acids: WASLKRLVTRRKRS (cluster of basic amino acids is underlined).

Response 3: We have performed additional experiments to assess the ability of peptides of different length spanning the ‘WASLKRLV’ sequence to inhibit interaction between AKAP79 1-153 and CaM. In line with the reviewer’s comment, we found that inclusion of the amino acids ‘RRKRS’ leads to an increase in potency indicated by a $K_i = 75 \pm 8$ nM (16-mer peptide, **Fig. 3d**, black line) compared to 170 ± 40 nM for an 11-mer peptide lacking these basic amino acids (**Fig. 3c**, red line). The sequence logo for the AKAP5 gene family (**Fig. 3b**) also shows that these amino acids are highly conserved across the AKAP5 gene family. We thank the reviewer for their insight on this point.

The authors performed cocrystallization experiments between CaM and the AKAP79 amino acids 77-92 comprising the 1-8-14 motif. It is a known fact that short CaM-target peptides do not always accurately reflect the binding behavior of longer peptides or the full length protein. One example is the abovementioned Munc13 protein where short peptides reflected a more compact formation of the CaM/Munc13 peptide complexes. But when studying longer peptides, it became apparent that the structure of the CaM/Munc13 complex is only correctly reflected if the additional hydrophobic anchor amino acid at position 26 is present (see publication attached). It is therefore crucial to use C-terminally elongated peptides containing all potential hydrophobic anchor residues. A CaM-binding motif in AKAP79 with an additional, more C-terminal, anchor residue would also be in agreement with the finding that CaM-binding is abolished when amino acid 78-86 are removed in AKAP79. Also, the additional hinge point at Leu-101 in AKAP79 would be in agreement with the “hot-spot” cross-link between Lys-94 in CaM and Lys-99 in AKAP79.

Response 4: The reviewer reasonably suggests that we consider whether additional hydrophobic elements lie downstream of the ‘WASLKRLV’ sequence, and proposes Leu101 as a potential additional hydrophobic anchor point. Taking W79_{AKAP79} as position 1 in the binding sequence, there is no secondary hydrophobic cluster within the first 30 amino acids equivalent to the sequence ‘LWF’ in Munc13 (position 26 is underlined). The best candidate – Leu101 – falls within the sequence ‘KPLEG’ which contains no other large aliphatic or aromatic amino acids. Furthermore, leucine is not strongly conserved at position 101 across the AKAP5 family (**Fig. 3b**). Nevertheless, we compared the potency of 26-mer peptides spanning AKAP79 77-102 to inhibit CaM – AKAP79 (1-153) interaction. Extension of the peptide C-terminus from position 96 to 101 led to only a modest increase in potency ($K_i = 22 \pm 1$ from 32 ± 2 nM, **Fig. 3d & e**). Mutating leucine101 to alanine in the 26-mer peptide had no significant effect on potency (**Fig. 3e**, $K_i = 24 \pm 2$ nM). In addition, experiments with Ca²⁺ binding-deficient mutants of CaM suggest that Ca²⁺ binding to the N-lobe is not required for interaction with AKAP79 (**Fig. 6e**, red line = CaM12, N-lobe deficient mutant). Therefore, we conclude that a secondary hydrophobic cluster is not

important for the interaction between AKAP79 and Ca²⁺/CaM.

I therefore suggest that the authors perform additional X-ray crystallography experiments with C-terminally elongated AKAP79 peptides, comprising Leu-101.

Response 5: We have indeed attempted to co-crystallise CaM with longer AKAP79 peptides, but crystals have not been forthcoming (unfortunately successful crystallization can never be guaranteed, particularly for complexes). It may be that longer peptides impede crystal packing in this case.

Additional comment on MS analysis: The authors should aim for recording MS/MS data with high resolution and high mass accuracy. In this work, mass tolerance was set to 0.3 Da (page 23) for cross-linked peptides, which will not allow an unambiguous identification of cross-linked products, especially for highly charged species.

Response 6: We thank the reviewer for this suggestion and agree that low mass tolerance is in principle advantageous for unambiguous identification in proteomic experiments. However, the workflow for the identification of cross-linked peptides used in this study – applying exactly the mass tolerances used in this study – has been successfully tested and applied numerous times. It is the first general workflow and software suite for the analysis and validation of crosslinks from large protein complexes by mass spectrometry (Leitner et al., Nature Protocols, 2014, PMID: 24356771; Rinner et al., Nature Methods, 2008, PMID:18327264) and it is widely-used and accepted in the field. It has been benchmarked and shown to be able to identify crosslinked peptides unambiguously, reproducibly and with high-confidence (Walzthoeni et al., Nature Methods, 2015 PMID: 26501516; Erzberger et al., Cell, 2014, PMID: 25171412). It has been applied to a large number of protein complexes and molecular machines - many much larger and more complicated than the complex investigated in this study (E.g. Greber et al., Cell, 2016, PMID:26709046; Kosinski, Science, 2016, PMID: 27081072; Petrovic et al., Cell, 2016, PMID:27881301) – for a recent review see for example Leitner et al., Trends Biochem Sci, 2016, PMID:26654279). We are therefore confident that the workflow and settings used in this study allowed for the trustworthy and unambiguous identification of cross-linked peptides.

Reviewer #3 (Remarks to the Author):

This manuscript details the very interesting finding of a novel, previously unseen, mode of calmodulin (CaM) - substrate binding. Given the importance of CaM in sensing intracellular changes in calcium concentration and using this to modulate the activities of a great many substrate proteins, this finding should prove of great interest. The manuscript is well-written and the work presented appears to be well done. I do have a couple of minor issues I would like to see addressed.

Response 1: We thank the reviewer for their encouraging comments.

1) The authors have measured CaM binding to calcineurin to be 30±12nM. This is at odds with published data that is in the low picomolar range (e.g. O'Donnell et al. (2011) Proteins 79:765 and Quintana et al. (2005) BBRC 334:674). The authors need to address this discrepancy.

Response 2: The reviewer has picked up on an important distinction that we overlooked in our original submission: half-maximal CaM binding to calcineurin occurs at a lower CaM concentration than required for half-maximal activation of the phosphatase. We have altered the text to clarify that the experiments in **figure 4** indicate that calcineurin activation is not required for association with the N-terminus of AKAP79 but we cannot rule out whether CaM *binding* to calcineurin is required (p. 12, lines 3-6). This however does not affect the key conclusion that the 'WASLKRLV' sequence in AKAP79 is required for the Ca²⁺-dependent calcineurin-AKAP79 interface.

Our calculated value of K_{CaM} for calcineurin activation is in line with previous publications (e.g. Fig. 1 of Quintana et al., 2005). Regarding known values for half-maximal binding of CaM to calcineurin. Our understanding is that there is some uncertainty in this figure with O'Donnell et al., 2011

and Quintana et al., 2005 reporting much tighter binding on the basis of very slow k_{off} rates compared to earlier reports (e.g., Perrino et al., 2002, Eur J Biochem 269: 3540-3548). However, it is clear that half-maximal CaM binding occurs at a lower CaM concentration than required for half-maximal calcineurin activation. We thank the reviewer for alerting to us to this important oversight in our original submission.

2) Why was the alpha-screen used to measure binding? There are multiple methods available that would not require coupling of the proteins to large beads, or fusion to other proteins, that could interfere with the measurements. Were the alpha-screen results validated against another method?

Response 3: We had hoped to take advantage of isothermal titration calorimetry but yields of purified AKAP79 were not sufficient to take advantage of this technique. An advantage of the alphascreen approach is that it is suited to high-throughput measurements. We have more effectively exploited this advantage with additional experiments included in our revised manuscript including the peptide ‘walk’ experiment presented in **figure 2c & d**. Other quantitative protein-protein interaction techniques that we considered also suffer from shortcomings. For example, surface plasmon resonance requires immobilization of one of the binding partners and is relatively expensive; and fluorescence-based approaches (e.g., Quintana et al., BBRC, 2005) require mutation of CaM to incorporate a cysteine residue and attachment of a bulky fluorophore that may affect target binding. In addition to existing crosslinking and crystallographic data that are consistent with our alphascreen measurements, we include additional pull-down experiments in our revised submission to validate the results of alphascreen experiments. Pull-down of AKAP79 variants from HEK cells using CaM sepharose and cAMP agarose (**Fig. 3a**) validates alphascreen-based peptide scan experiments presented in **figure 2c & d**. Furthermore, the inability of purified AKAP79 W79A to bind CaM sepharose is consistent with alphascreen-based peptide disruption experiments presented in **figure 5c** that suggest this amino acid is a key contact point.

3) Figure 3a shows the locations of the isoleucine to alanine mutations made in the calcineurin CaM binding region. Although the figure is adequate to show the locations of the mutations, the full sequence of the CaM binding region is not shown, which I find a little misleading.

Response 4: We have extended the sequence showing the CaM binding site in calcineurin presented in **figure 4a**. We thank the reviewer for their perceptive comments.

Reviewers' comments:

Reviewer #1 (Remarks to the Author):

The authors have taken the comments thoroughly and the quality of the manuscript has been significantly improved. I have no further comments and congratulate the authors on an excellent paper.

Reviewer #2 (Remarks to the Author):

I find this version of the manuscript to be improved compared to the previous version. The additional experiments that were conducted by the authors are helpful and enhance the overall quality of this paper. However, a crystal structure of the complex of a longer AKAP79 peptide with CaM is still missing.

One point however has to be addressed (and I feel very strongly about this) is the quality of MS/MS data. It is crucial to record both MS and MS/MS data with high mass measurement accuracy to allow an unambiguous identification of crosslinks. A paper (Anal Chem 2017) has recently been published addressing this issue. Merely stating that a number of papers have been published using MS/MS data with low mass accuracy does not imply that the method used is correct.

In fact, this is a highly dangerous development that many papers in the field of structural biology are now getting published where MS data do NOT fulfill the necessary basic requirements for an unambiguous assignment of crosslinks.

In order to make this paper solid, data should be recorded on a mass spectrometer with high mass accuracy and high resolution (both on MS and MS/MS level), which would be the best option. At least, the authors should clearly state in their paper that their MS/MS data are not of optimal quality and show that they are aware of this fundamental problem.

Responses to reviewer comments

Title “Crosslinking and crystallography reveal a novel type of calmodulin interaction motif in AKAP79/150”

Tracking code: NCOMMS-17-02050-A

Authors: Patel, Stengel, Aebersold & Gold

Reviewer #1 (Remarks to the Author):

The authors have taken the comments thoroughly and the quality of the manuscript has been significantly improved. I have no further comments and congratulate the authors on an excellent paper.

Response 1: We thank the reviewer for their generous comments.

Reviewer #2 (Remarks to the Author):

I find this version of the manuscript to be improved compared to the previous version. The additional experiments that were conducted by the authors are helpful and enhance the overall quality of this paper. However, a crystal structure of the complex of a longer AKAP79 peptide with CaM is still missing.

Response 2: We are pleased to hear that the reviewer finds our manuscript to be improved. We acknowledge that ideally we could include a crystal structure of a complex containing a larger fragment of AKAP79. As mentioned in our previous author response letter, we have attempted to crystallize co-complexes containing longer peptides without success. The nature of crystallography is that some complexes do not readily crystallize. The structure of calmodulin – AKAP79 (77-92) that we have solved itself required extensive crystal optimization before high quality diffraction data was obtained. In this case, the structure was solved more than a year after the first crystal hit was identified.

One point however has to be addressed (and I feel very strongly about this) is the quality of MS/MS data. It is crucial to record both MS and MS/MS data with high mass measurement accuracy to allow an unambiguous identification of crosslinks. A paper (Anal Chem 2017) has recently been published addressing this issue. Merely stating that a number of papers have been published using MS/MS data with low mass accuracy does not imply that the method used is correct.

In fact, this is a highly dangerous development that many papers in the field of structural biology are now getting published where MS data do NOT fulfill the necessary basic requirements for an unambiguous assignment of crosslinks.

In order to make this paper solid, data should be recorded on a mass spectrometer with high mass accuracy and high resolution (both on MS and MS/MS level), which would be the best option. At least, the authors should clearly state in their paper that their MS/MS data are not of optimal quality and show that they are aware of this fundamental problem.

Response 3: We assume that the reviewer refers to the interesting paper by Iacobucci & Sinz

(Analytical Chemistry, 2017, PMID 28723100). We agree that the points discussed in this paper are important to consider for a successful XL-MS experiment, and note that our workflow adheres to these points with the sole exception of the suggested ppm cutoff for MS/MS. In some instances we apply more stringent criteria than recommended, for example, we apply a minimum length of 5 amino acids for crosslinked peptides to assure thorough sequencing of identified crosslinks.

Regarding the specific issue of high mass accuracy and high resolution. Our experiments were performed using an Orbitrap Elite mass spectrometer, which is “a mass spectrometer with high mass accuracy and high resolution (both on MS and MS/MS level)”. In our set-up, we use an ion trap to measure and detect MS/MS spectra. In our opinion, there are important experimental reasons for using the MS/MS settings that we applied. The combined set-up of high-resolution MS in the Orbitrap and lower resolution MS/MS in the ion trap enables quicker measurement cycles. While this set-up achieves lower mass accuracy, in particular in comparison to an Orbitrap mass analyzer, this is outweighed by the ability to sample a larger part of the total ion current. More importantly, our combined set-up also exhibits better sensitivity, thereby helping to identify crosslinked peptides that are very often of low abundance. The parameter set that we applied has been optimized over many years and validated on multiple occasions. In our study, interlinks detected between AKAP79 and calmodulin are highly consistent with results from other approaches including the peptide ‘walk’ experiments presented in **Figure 2c**.

However, in order to state more clearly that different workflows for crosslink assignment are present in the community, we have included an additional sentence in the methods section as follows (*p. 26, lines 20-21*):

“Fragment ions were analyzed in an ion trap, trading lower mass accuracy for increased sensitivity and measuring speed.”